# DOMAIN-AWARE TENSOR NETWORK STRUCTURE SEARCH

## ABSTRACT

Tensor networks (TNs) provide efficient representations of high-dimensional data, yet identification of the optimal TN structures, the so called tensor network structure search (TN-SS) problem, remains a challenge. Current state-of-the-art (SOTA) algorithms solve TN-SS as a purely numerical optimization problem and require extensive function evaluations, which is prohibitive for real-world applications. In addition, existing methods ignore the valuable domain information inherent in real-world tensor data and lack transparency in their identified TN structures. To this end, we propose a novel TN-SS framework, termed the tnLLM, which incorporates domain information about the data and harnesses the reasoning capabilities of large language models (LLMs) to *directly* predict suitable TN structures. The proposed framework involves a domain-aware prompting pipeline which instructs the LLM to infer suitable TN structures based on the real-world relationships between tensor modes. In this way, our approach is capable of not only iteratively optimizing the objective function, but also generating domain-aware explanations for the identified structures. Experimental results demonstrate that tnLLM achieves comparable TN-SS objective function values with much fewer function evaluations compared to SOTA algorithms. Furthermore, we demonstrate that the LLM-enabled domain information can be used to find good initializations in the search space for sampling-based SOTA methods to accelerate their convergence while preserving theoretical performance guarantees. Our code is included in the supplementary materials.

## 1 INTRODUCTION

The exponential increase in the volume and richness of available data has led to the widespread use of multi-way arrays, often represented as higher-order tensors. Tensor network decomposition methods aim to represent higher-order tensors in "super-compressed" formats through smaller-sized components, by effectively capturing cross-modal latent patterns and correlations. These methods have been applied across various fields, including machine learning (Zhe et al., 2015; Cichocki et al., 2016; Malik, 2021), signal processing (Zheng et al., 2023), computer vision (Shakeri & Zhang, 2019; Yamamoto et al., 2022), and quantum physics (Orús, 2019; Felser et al., 2021). The success of tensor network decomposition techniques is closely linked to their ability to mitigate the "curse of dimensionality", which is achieved by decomposing higher-order data into lower-order factors.

However, tensor network (TN) practitioners face significant challenges related to choosing the most appropriate TN structure, the so called *tensor network structure search (TN-SS)* problem, which has been proven to be NP-hard (Hillar & Lim, 2013; Li et al., 2023). TN-SS involves determining the optimal TN parameters, such as TN ranks, TN topology, and TN mode permutations (Li et al., 2022).

Existing TN-SS methods solve TN-SS as a purely numerical optimization problem and include approaches such as Bayesian inference (Zeng et al., 2024b), greedy algorithm (Hashemizadeh et al., 2020), discrete optimization (Li et al., 2022; 2023), and continuous optimization (Zheng et al., 2024). To date, sampling-based methods (Li & Sun, 2020; Li et al., 2022; 2023; Zeng et al., 2024a), whose workflows are illustrated in Figure 1a and 1b, have demonstrated the best performance in addressing the TN-SS problem.

However, these methods require large number of evaluations (of the training and testing data), are prone to getting stuck in local minima, and lack transparency in their found structure-related parameters (Li et al., 2023). Critically, a large number of evaluations needed to optimize the objective

function leads to a high computational cost. We hypothesize that these limitations arise from failing to exploit the rich domain information inherent in real-world tensors, such as the mode information. To this end, we ask ourselves:

*How can we utilize the intrinsic domain information in tensor data to significantly reduce the number of evaluations required to solve the TN-SS problem, while providing domain-aware explanations for the identified TN solutions?*

To address this question, we propose a domain-aware large language model (LLM)-guided TN-SS framework, termed tnLLM. Within this framework, an LLM is utilized to initialize the TN structure based on domain information about the relationships between tensor modes. Then, the reasoning capabilities of the LLM are used to navigate the search space effectively, in order to achieve good optimization of the objective function with very few evaluations.

Our proposed framework is found to achieve significant speed-ups over current state-of-the-art (SOTA) methods in terms of the number of evaluations, due to its ability to find good TN structure initializations through the use of domain information. Moreover, tnLLM generates practically meaningful explanations for the TN solutions, thus offering transparency in the identified structures. This is particularly beneficial for tensor practitioners who lack deep expertise in a specific data domain, as it enables them to both comprehend the interactions between tensor modes and trust the discovered TN structures. It also allows the identified structures to be verified by domain experts.

To evaluate the effectiveness of tnLLM, we compare its performance against SOTA TN-SS algorithms on real-world tensor datasets of order-$3$, $4$, and $5$. The experimental results demonstrate that tnLLM delivers performance comparable to current SOTA methods, while requiring significantly fewer function evalua-

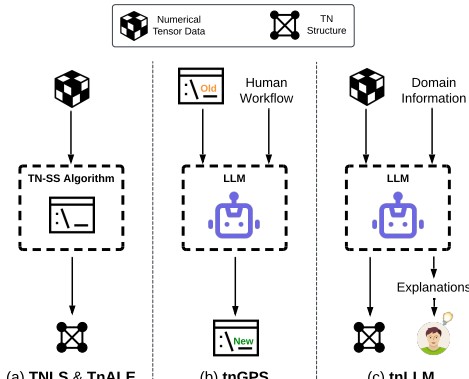

Figure 1: Comparison of SOTA tensor network structure search (TN-SS) methods. (a) TN-SS algorithms with theoretical guarantees: TNLS (Li et al., 2022) & TnALE (Li et al., 2023). (b) tnGPS (Zeng et al., 2024a), a prompting pipeline which uses LLMs to generate new TN-SS algorithms. (c) tnLLM (ours), which uses domain information about the tensor data and LLM reasoning to solve the TN-SS problem and generate explanations for the identified TN structure.

tions and providing domain-aware explanations for the identified TN structures. Moreover, we constructed a hybrid algorithm to combine the speed-up benefits of tnLLM with the theoretical guarantees of existing sampling-based approaches. The main contributions of this work are:

- We propose tnLLM, a novel domain-aware LLM-guided TN-SS framework, which achieves performance on par with SOTA methods while requiring much fewer evaluations;

- To the best of our knowledge, our framework is the first to utilize domain information inherent in real-world tensor data to address the TN-SS problem. This enables the generation of domain-aware explanations that allows practitioners to verify the identified TN structures.

## 1.1 RELATED WORK

**Tensor network structure search (TN-SS).** Compared to traditional tensor decompositions (Hitchcock, 1927; Tucker, 1966; Oseledets, 2011; Zhao et al., 2016; Wu et al., 2022), which have pre-defined tensor network structures, the TN-SS problem focuses on finding custom tensor networks, which have been shown to achieve higher parameter efficiency and are an important paradigm of tensor decompositions (Li et al., 2022; 2023; Li & Sun, 2020; Zeng et al., 2024a). Various approaches have been proposed to address the TN-SS problem, including Bayesian inference (Zeng et al., 2024b), greedy algorithm (Hashemizadeh et al., 2020), program synthesis (Guo et al., 2025), and continuous optimization (Zheng et al., 2024). Sampling-based methods (Li et al., 2023; 2022; Li & Sun, 2020; Zeng et al., 2024a), which fall under discrete optimization methods, have demonstrated superior performance compared to other approaches in addressing the TN-SS problem. Among these, TNLS

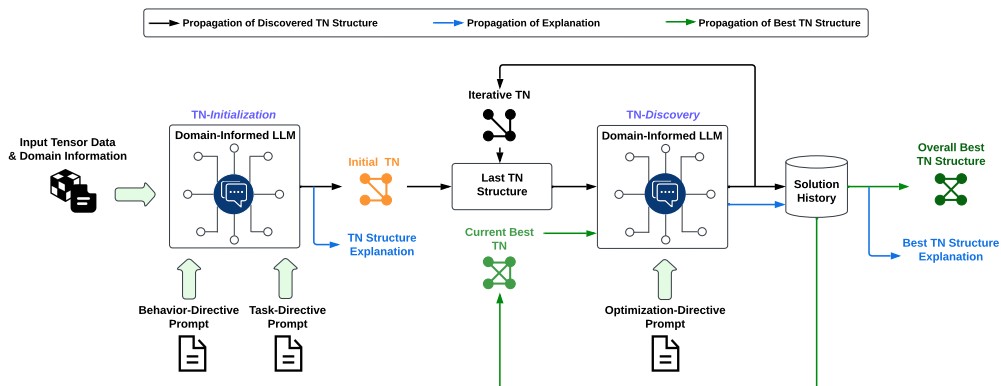

Figure 2: The workflow of the proposed tnLLM framework consists of two key stages commonly adopted by SOTA TN-SS methods (Li et al., 2022; 2023; Zeng et al., 2024a): *TN-initialization* and *TN-discovery*. The proposed tnLLM efficiently utilises the domain information in tensor data and uses an LLM to guide these two stages.

(Li et al., 2022) and TnALE (Li et al., 2023) achieve state-of-the-art (SOTA) performance, with TnALE accelerating TNLS by slightly reducing the number of evaluations required to optimize the objective function. While both TNLS and TnALE adopt a "local-search" scheme, this acceleration is achieved by the alternating variable updates proposed in TnALE. However, these methods still face challenges, such as a large number of function evaluations required to converge, difficulty in balancing exploration and exploitation, and a lack of transparency in the found structures.

More recently, tnGPS (Zeng et al., 2024a) has emerged as an approach that uses LLMs to generate sampling-based TN-SS algorithms, demonstrating performance comparable to SOTA methods. Despite being closest to our work, tnGPS focuses on generating TN-SS algorithms based on sampling-based heuristics and does not incorporate any domain information in the tensor data. As such, tnGPS suffers from the same limitations as existing sampling-based methods, including a lack of explainability in the identified structure-related parameters.

To this end, our work proposes to harness the domain information in tensor data to solve the TN-SS problem. We achieve this by using the domain knowledge and the inherent reasoning capabilities of LLMs to *directly* infer tensor network structures. Facilitated by our carefully designed prompting pipeline (see Section 3), the proposed framework is shown not only to require significantly fewer evaluations, but also to produce domain-aware and verifiable solutions to the TN-SS problem.

**Reasoning with large language models.** The rise of transformer-based LLMs, pretrained on vast text corpora, has demonstrated a remarkable capacity for "reasoning" (Wei et al., 2022a). This reasoning ability is further enhanced when LLMs are guided by task-specific prompting strategies, such as chain-of thought (Wei et al., 2022b; Suzgun et al., 2023; Kojima et al., 2024). These strategies enable LLMs to generate text effectively for tasks such as arithmetic reasoning, optimization (Yang et al., 2024), and factual knowledge retrieval, leading to exceptional performance in complex question-answering and tasks requiring real-world knowledge (Liévin et al., 2023; Singhal et al., 2023). These findings suggest that, through pretraining on diverse textual data, LLMs encode rich knowledge about real-world relationships, which they can effectively leverage to perform various downstream tasks (Choi et al., 2022). In this work, we utilize the real-world domain knowledge and "reasoning" capabilities of LLMs to directly infer suitable TN structures.

## 2 PRELIMINARIES

### 2.1 TENSOR NETWORK STRUCTURE SEARCH (TN-SS)

We first provide the definition of TN-SS through its application in higher-order data decomposition. An optimal solution to the TN-SS problem aims to find the best trade-off between identifying the most compressed TN structure while preserving the expressivity of the TN (Li & Zhao, 2021).

---

**Algorithm 1:** Sampling-based Algorithms for TN-SS (Li et al., 2022; 2023; Zeng et al., 2024a)

1: **Initialize:**
2: $N_{Iter}$           ▷ *Maximum number of iterations*
3: $N_{Sample}$        ▷ *Number of samples per iteration*
4: $P \leftarrow []$         ▷ *Historical TN structures*
5: $H \leftarrow []$         ▷ *Discovered TN structures in each iter.*
6: $\mathcal{F}(\cdot) \leftarrow$ Equation (1)         ▷ *Objective function*
7: $(G, \mathbf{r}) = $ *Initial TN Structure*         ▷ *TN-initialization*
8:
9: **Algorithm:**
10: **for** $n = 1$ **to** $N_{Iter}$ **do**
11:     $H \leftarrow N_{Sample}$ TN structures sampled in the neighborhood of $(G, \mathbf{r})$.     ▷ *TN-discovery*
12:     $P \leftarrow P \cup H$
13:     **if** $\exists (\hat{G}, \hat{\mathbf{r}}) \in P$ such that $\mathcal{F}(\hat{G}, \hat{\mathbf{r}}) < \mathcal{F}(G, \mathbf{r})$ **then**
14:        $(G, \mathbf{r}) \leftarrow (\hat{G}, \hat{\mathbf{r}})$.
15:     **if** Converged **then**
16:        **return** $(G, \mathbf{r})$
17: **Output:** $(G, \mathbf{r})$

---

Let $\mathcal{X} \in \mathbb{R}^{I_1 \times I_2 \times \cdots \times I_N}$ be a non-zero order-$N$ tensor, with its Frobenius norm as $\|\mathcal{X}\|_F$. Tensor networks can be represented using the graphical notation (Orús, 2014; Ye & Lim, 2018), whereby a tensor network $G$ is represented using a set of $N$ vertices, $V$, and a set of edges, $E$, i.e., $G = (V, E)$. Each vertex represents a decomposed core tensor, and the closed edges between two core tensors are generalized higher-order matrix multiplications, termed tensor contractions (Cichocki et al., 2016). Closed edges have assigned *TN-ranks*, $\mathbf{r} \in \mathbb{Z}_+^E$, which indicate the degree of connectivity between different pairs of connected vertices. Therefore, the properties of a TN structure can be fully expressed by the combination $(G, \mathbf{r})$.

Similar to SOTA TN-SS methods (Li & Sun, 2020; Li et al., 2023; 2022; Zeng et al., 2024a), the discrete optimization problem of TN-SS is formalized as a minimization of the objective function, which is a linear sum of the complexity of the TN structure (e.g., compression rate) and the TN expressivity (e.g., approximation error), and is given by

$$\min_{(G, \mathbf{r})} \ln \left( \phi(G, \mathbf{r}) + \frac{\lambda}{L} \min_{\{\mathcal{V}_{l,i}\}_{i=1}^N} \sum_{l=1}^{L} \frac{\left\| \mathcal{X}_l - TNC\left( \{\mathcal{V}_{l,i}\}_{i=1}^N ; (G, \mathbf{r}) \right) \right\|_F}{\|\mathcal{X}_l\|_F} \right) \tag{1}$$

where $L$ represents the number of tensor samples in the dataset, and the first term $\phi(G, \mathbf{r})$ measures the TN structure complexity. The second term in Equation (1) measures the expressivity of the TN through the relative squared error (RSE) between the original tensors $\{\mathcal{X}_l\}_{l=1}^L$ and their TN approximations $\left\{ TNC\left( \{\mathcal{V}_{l,i}\}_{i=1}^N ; (G, \mathbf{r}) \right) \right\}_{l=1}^L$, where $TNC(\cdot)$ stands for the tensor contraction operation of the entire tensor network. The pair $(G, \mathbf{r})$ characterizes how the vertices, $\{\mathcal{V}_{l,i}\}_{i=1}^N$, are contracted together to approximate the original tensor $\mathcal{X}_l$. The coefficient $\lambda$ is a positive non-zero scaling factor which balances the trade-off between model complexity and model expressivity. Note that (Li & Sun, 2020; Ye & Lim, 2018) pointed out that the TN-SS problem is conveniently equivalent to the TN rank search problem of a fully connected TN (Zheng et al., 2021).

## 2.2 SOTA SAMPLING-BASED ALGORITHMS FOR TN-SS

Algorithm 1 summarizes the current SOTA algorithms for TN-SS. They follow a three-step search process: *TN-initialization* → *TN-discovery* via sampling in the search neighborhood → Updating the center of the search neighborhood. Existing methods ignore the inherent domain information in real-world tensor data, which calls for the development of a framework that can effectively utilize domain knowledge in *TN-initialization* and *TN-discovery* to improve performance.

To this end, our proposed framework introduces domain-aware LLM-guided *TN-initialization* and *TN-discovery*. By doing so, it addresses the limitations of existing methods, such as high computational costs caused by the large number of evaluations ($N_{Iter} \times N_{Sample}$) required and the tendency to get stuck in local minima due to difficulties in balancing the exploration-exploitation trade-off. Importantly, by incorporating domain information, our tnLLM framework provides practically meaningful explanations for the identified TN structures, a feature absent in current approaches.

# 3 TNLLM: A DOMAIN-AWARE FRAMEWORK FOR SOLVING TN-SS

This section presents tnLLM, a domain-aware LLM-guided framework that efficiently solves the TN-SS problem with very few evaluations and verifiable solution explanations. We detail the role of each component in the prompting pipeline and the guidelines followed by each prompt.

**Workflow of tnLLM.** Figure 2 illustrates the workflow of the proposed tnLLM framework. The "Behavior-directive" prompt is used to supply the problem specification and guide the general-purpose LLM into a model tailored for solving the TN-SS problem. Inspired by current sampling-based SOTA algorithms, this domain-informed LLM is then employed for LLM-guided *TN-initialization* and *TN-exploration*.

During the *TN-initialization* stage, the "Task-directive" prompt is used to propose a strong initial TN structure by using the provided domain information. In the *TN-discovery* stage, the last and best-identified TN structures are used in conjunction with the "Optimization-directive" prompt to guide the LLM in navigating the search space effectively. The identified TN structures are fed into an objective function evaluation program to obtain the objective function values. This iterative process leads to the refinement of the TN structure and yields improved objective function values over successive evaluations.

**Prompting pipeline.** The interactions between the carefully designed user prompts and the LLM assistant follow an automated structured dialogue (OpenAI, 2023; Touvron et al., 2023). This dialogue-based approach is intuitive for generating stepwise conversational reasoning and is particularly useful for iterative tasks such as the TN-SS problem. To further improve the ability of the LLM to perform complex reasoning, we employ the chain-of-thought prompting (Wei et al., 2022b). This prompting strategy breaks down complex tasks into a series of intermediate reasoning steps and thus enables the model to solve the problem by addressing smaller, simpler sub-tasks sequentially. In the proposed tnLLM framework, three distinctly purposed prompts are employed, each serving a unique role. The full prompts are given in Appendix B.

**"Behavior-directive" prompt.** This is the system message in the dialogue. Its primary objective is to establish the role of the LLM as an expert in the TN-SS problem (orange section of Figure 3a) and guide its behavior throughout the dialogue (green section of Figure 3a). Additionally, it sets the context by outlining the TN-SS problem and the objective function (blue section of Figure 3a).

**"Task-directive" prompt.** This is the first non-system prompt supplied to the LLM and serves as the basis for LLM-guided *TN-initialization*. It is designed to thoroughly explain the domain information about the tensor structure, including details such as the number of modes and the specific information about each mode (orange section of Figure 3b). Furthermore, it guides the model's thought process and ensures that its responses align with the task requirements by instructing it to reason step-by-step and utilize domain knowledge (blue section of Figure 3b).

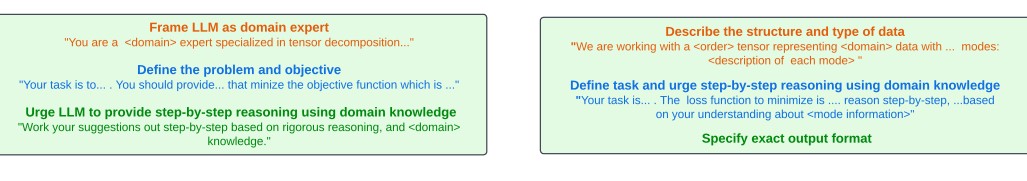

(a) "Behavior-directive" prompt                    (b) "Task-directive" prompt

Figure 3: Structure and components of (a) the "Behavior-directive" prompt, which frames the LLM's role as a domain expert, and (b) the "Task-directive" prompt for LLM-guided *TN-initialization*.

**"Optimization-directive" prompt.** This is the iterative prompt supplied to the LLM in all subsequent evaluations and serves as the basis for LLM-guided *TN-discovery*. Its purpose is to guide the LLM to efficiently navigate the search space by incorporating context from the best and previous evaluations (orange section of Figure 4). The prompt explains how refinements to the identified TN structure influence the objective function value and instructs the model to leverage step-by-step

reasoning and domain knowledge to optimize the objective function (blue section of Figure 4). At the same time, it encourages the LLM to both *explore* and *exploit* new solutions to reduce the likelihood of the method being stuck in local minima (purple section of Figure 4).

**Efficient and effective search space navigation.**   A critical factor in the performance of tnLLM, both in terms of evaluations and objective function value, is the incorporation of memory (orange section of Figure 4) and the encouragement of search space "exploration and exploitation" (purple section of Figure 4) in the "Optimization-directive" prompt. By including the best and most recent TN solutions, along with their respective objective function values and the domain information about mode interactions, the LLM gains a better understanding of the search space and can make more informed decisions. Additionally, by encouraging the model to explore new solutions when necessary and exploit good solutions already found assists the optimization process, by reducing the possibility of the model getting stuck in local minima, a common caveat of sampling-based methods.

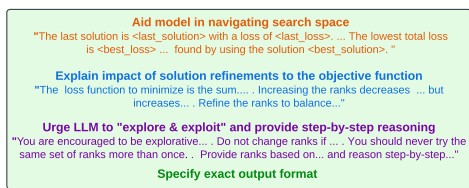

Figure 4: Structure and components of the "Optimization-directive" prompt used for LLM-guided *TN-discovery*.

**Output format specification.**   As observed in Figures 3b and 4, in addition to the goal-oriented components of each prompt, an essential feature (highlighted in green) of both the "Task-directive" and "Optimization-directive" prompts is the specification of the exact output format of the TN-SS solution. This includes defining both the sequence in which the LLM should present its response and the precise format of the TN-SS solution. To meet these requirements, the prompt shown in Figure 5 was developed through experimentation and a trial-and-error process with the LLM. This step serves to eliminate "hallucinations" in the LLM output and enables a *fully automated* prompting pipeline without the need for any human intervention during the entire iterative optimization process.

*Remark* 3.1. Both tnGPS and tnLLM use LLMs to solve the TN-SS problem, however, the two approaches are fundamentally different. tnGPS uses the LLM to generate new *sampling-based TN-SS algorithms* by reviewing existing algorithms, causing the discovered methods to inherit similar limitations. In contrast, tnLLM uses the LLM to *directly infer TN structures* to solve the TN-SS problem by incorporating the rich domain information in real-world tensor data. In turn, this allows tnLLM to provide explanations for the TN solutions, thereby adding transparency to the identified TN structures.

```
... Output format:
Reasoning: Reason about the intrinsic interactions between every pair of modes
based on your understanding of <domain> data.
Solution: <Exact solution format>.
End the output message with <description of solution format>.
"""
```

Figure 5: The prompt used to specify the LLM output format.

## 4   EXPERIMENTAL RESULTS

In this section, the performance of tnLLM was evaluated against the SOTA sampling-based methods in tensor decomposition tasks in terms of the number of evaluations and objective function value. Our results demonstrate that tnLLM achieves comparable objective function values while delivering significant speed-ups. Moreover, we demonstrate its ability to generate domain-consistent explanations for the identified TN structures. Finally, an ablation study was conducted to evaluate the effectiveness of domain information and assess the framework's robustness across different LLMs.

**Data preparation.**   Given the flexibility of our method to handle tensor data of any order, we evaluated its performance on three types of tensor data with varying sizes and dimensionalities across different domains. In particular, we tested on datasets of order-3 RGB images and order-4 RGB videos. We also curated a completely new order-5 financial time-series dataset of 142 tensors to ensure that it was not included in the training data of LLMs. This time-series tensor dataset, to the best of our knowledge, is the largest dataset in terms of number of samples ever considered in the TN-SS problem. All entries were standardized to values in $[0, 1]$, with $80\%$ of each dataset used for training, and the remaining $20\%$ for testing. More details about the data can be found in Appendix C.

Table 1: Performance comparison across different datasets. The values on the left give the lowest training and corresponding testing objective function values. The values in [square brackets] give the number of evaluations required to first achieve the best training objective function value. For robustness assessment of tnLLM, we report the average and standard deviation of both the objective function value and the number of evaluations across 5 independent runs. For both metrics, a lower value is better. The best values are denoted in bold. The second best values are underlined.

| Data Type | | TNLS | | TnALE | | tnGPS | | tnLLM (Ours) | |
|---|---|---|---|---|---|---|---|---|---|
| **Images** | Train | -0.66 | [114] | -0.65 | [81] | -0.66 | [438] | -0.63±0.01 | [**4.0**±1.9] |
| | Test | -0.47 | | -0.46 | | -0.44 | | **-0.48**±0.05 | |
| **Videos** | Train | -1.64 | [484] | -1.66 | [254] | -1.65 | [175] | -1.63±0.01 | [**6.2**±3.5] |
| | Test | **-1.72** | | **-1.72** | | -1.71 | | -1.70±0.02 | |
| **Time-series** | Train | -0.45 | [218] | -0.47 | [177] | -0.39 | [38] | -0.42±0.02 | [**5.6**±4.4] |
| | Test | -0.43 | | **-0.47** | | -0.40 | | -0.41±0.02 | |

**Settings of tnLLM.** In Equation (1), we set $\lambda = 10$ and used the same compression ratio function, $\phi$, as in previous sampling-based methods (Li & Sun, 2020), defined as the ratio between the number of parameters in the compressed TN format and the original tensor. The maximum number of evaluations was set to 500 for the images and videos datasets, and 250 for the financial time-series dataset. An early stopping criterion with a patience of 5 was applied. For all experiments, the LLM model GPT-4o (`gpt-4o-2024-08-06`) (OpenAI, 2024) was used, with temperature set at 0.2.

**Implementation details.** In all experiments, we also implemented the three SOTA sampling-based TN-SS algorithms, namely TNLS (Li et al., 2022), TnALE (Li et al., 2023) and tnGPS (Zeng et al., 2024a), and accelerated them with GPUs. Since the vanilla TNLS is designed to search only for the permutation of a TN, we extended it to fit the settings of TN-SS. For fair comparisons, all baseline methods were evaluated using the same objective function and maximum number of evaluations, with one evaluation defined as a single pass through the entire training and testing dataset. The full implementation details are provided in Appendix D.

**Numerical results.** We ran tnLLM five times and examined the mean and standard deviation for both the objective function values and the number of evaluations to demonstrate its *robustness*. Observe from Table 1 that tnLLM achieved performance on par with SOTA sampling-based algorithms in both training and test objectives across all data types, as measured by the objective function. Importantly, tnLLM minimizes the objective function with significantly fewer evaluations, requiring up to $78\times$ fewer evaluations than TNLS, $41\times$ fewer than TnALE, and $110\times$ fewer than tnGPS. Consequently, tnLLM achieves runtime reductions of up to 98.3% compared to TNLS, 97.7% compared to TnALE, and 98.1% compared to tnGPS, even after accounting for LLM inference. The full runtime comparisons are provided in Appendix E. Furthermore, in Table 2, we report the best compression rates (number of entries in the original tensor/ number of entries in its TN compressed format) achieved by all methods for a given test approximation error threshold across three datasets (0.02 for Images and 0.01 for Videos and Time-Series datasets). The approximation error is calculated as $\frac{1}{L}\sum_{l=1}^{L}\frac{\left\|\mathcal{X}_l - TNC\left(\{\mathcal{V}_{l,i}\}_{i=1}^{N};(G,\mathbf{r})\right)\right\|_F}{\|\mathcal{X}_l\|_F}$. An error threshold of 0.01 can be interpreted as almost a 'perfect' reconstruction for standardized tensors. We use 0.02 as the test approximation error threshold for the Images dataset, as none of the methods achieved an error of 0.01. Observe from Table 2, that the proposed tnLLM achieves the best compression rate in two of the three datasets.

Table 2: Comparison of the best compression rates (No. entries in the original tensors / No. entries in the compressed TN formats) achieved by TNLS, TnALE, tnGPS, and tnLLM. Results are reported achieving a test approximation error below the threshold of 0.02 for Images and 0.01 for Videos and Time-Series. Bold values indicate the best compression performance. The second best are underlined. "Failed" means that the method failed to achieve an approximation error below the threshold.

| Dataset | TNLS | TnALE | tnGPS | tnLLM |
|---|---|---|---|---|
| Images | 2.54 | 2.25 | 2.47 | **2.63** |
| Videos | 12.38 | **12.67** | 11.52 | 11.22 |
| Time-Series | 1.47 | Failed | 1.49 | **1.60** |

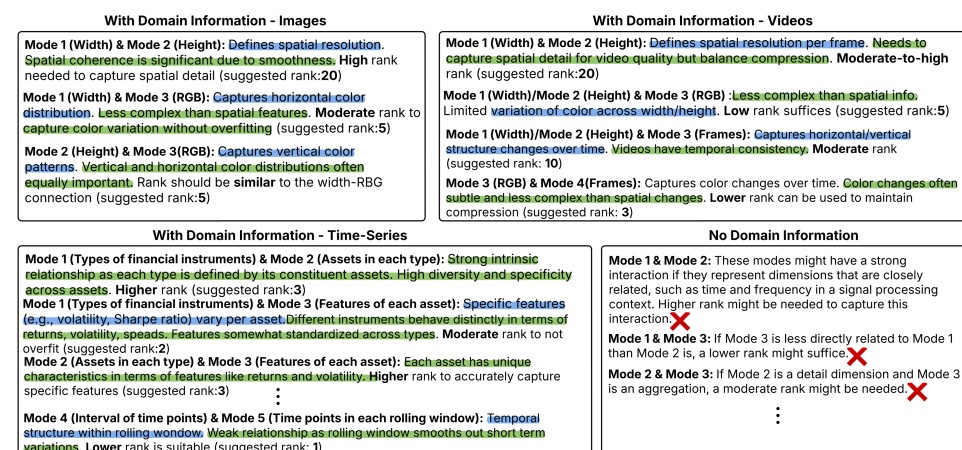

Figure 6: Summarized generated explanations of tnLLM for the initialized TN structure, with and without the incorporation of domain information in the prompts.

**Domain-aware explanations.** To assess the domain-relevance of the identified TNs, we manually verified the explanations provided for the identified TN structures in all three datasets tested. The explanations of the tnLLM model for the initialized TN structure, with and without domain knowledge, are provided in Figure 6. Observe that without domain information, although the model is inherently capable of reasoning, its lack of domain information about the modes in the real-world tensors causes it to default to random assumptions, likely influenced by its pre-training data, such as assuming a mode as resembling time or frequency. As a result, the explanations are entirely incorrect from a practical perspective and are therefore ineffective in solving TN-SS.

In contrast, incorporating domain information enables the model to identify relationships between modes (highlighted in blue) and, more importantly, relate them to real-world principles in vision and finance (highlighted in green). The model also shows consistent reasoning in its rank suggestions: when describing a rank as low, medium, or high, it quantitatively selects a value that is coherent relative to other ranks within the same tensor. Moreover, it adjusts these values based on comparisons across different mode pairs, for example by assigning similar ranks to the Height–RGB and Width–RGB modes in the images dataset, and a lower rank to the RGB–Frames mode in the videos dataset. This demonstrates the model's understanding of tensor mode relationships based on domain information and supports the validity of the identified TN structure. Consequently, tnLLM offers explanations that are practically useful in helping domain experts to verify and trust the discovered structures. Summarized explanations across three runs for all datasets are provided in Appendix F.

## 4.1 ACCELERATING SOTA METHODS WITH TNLLM

In the first part of Section 4, we have demonstrated that tnLLM achieves comparable objective function values compared to SOTA sampling-based algorithms, while generating domain-aware explanations for the identified TN structures and significantly reducing the required number of function evaluations. However, due to the black-box nature of LLMs, no theoretical analysis can be provided for the evaluation efficiency of tnLLM, in contrast to TNLS and TnALE.

At the same time, while sampling-based methods follow a "local-search" scheme within a neighborhood, there is no guarantee on the minimum number of evaluations required to find a "good" neighborhood. Moreover, poor initialization significantly increases the number of evaluations needed. To address this, we construct a hybrid algorithm that first runs tnLLM for 10 evaluations to leverage its domain knowledge and reasoning capabilities in order to identify a strong initialization point in the "global-search" stage. Sampling-based methods then perform "local-search" in the identified neighborhood, combining the speed-up benefits of tnLLM with their theoretical guarantees.

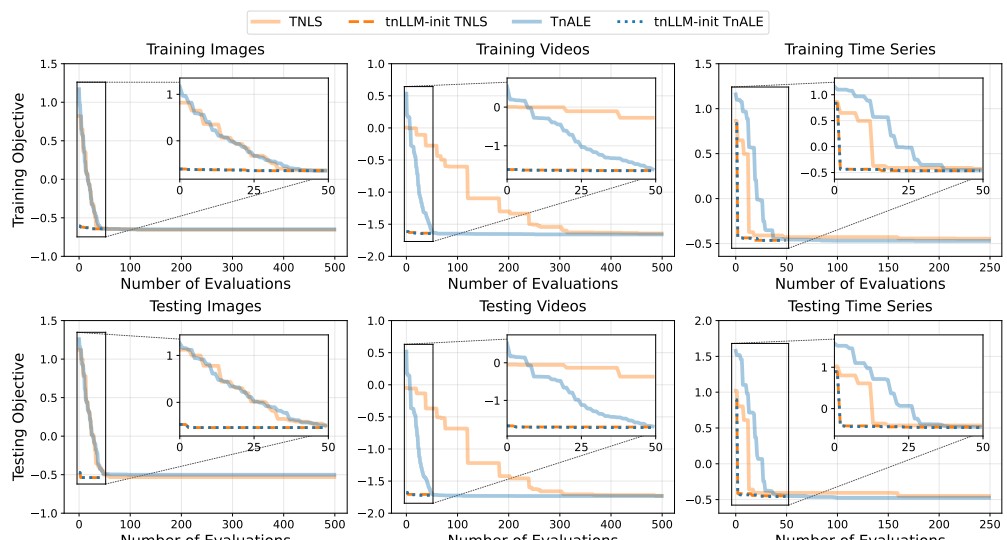

Figure 7: Experimental results across all 3 datasets show that both TNLS and TnALE are significantly accelerated with an initial center of search neighborhood found by tnLLM. We ran TNLS and TnALE with tnLLM initialization in all 3 datasets over 50 evaluations, which *include* the plotted 10 evaluations performed by tnLLM. Vanilla TNLS and TnALE performed 500 evaluations for both the images and videos datasets and 250 evaluations for the financial time-series dataset if not converged.

**Numerical results.** Figure 7 demonstrates the performance difference of TNLS and TnALE with and without a tnLLM-initialized structure. The "global-search" capability of tnLLM significantly sped up the iterative minimization process. During "local-search", both TNLS and TnALE further improved upon the structures found in "global-search" by tnLLM. Overall, the tnLLM-initialized algorithms achieved nearly identical objective function values with up to $23\times$ fewer evaluations compared to vanilla TNLS, and up to $13\times$ fewer evaluations compared to vanilla TnALE. The full numerical performance comparison is provided in Appendix G.

## 4.2 ABLATION STUDY

**Removal of domain information.** To assess the effectiveness of injecting domain information into the TN-SS problem, we removed the carefully designed structured prompts of the tnLLM framework, equivalent to using the LLM to solve the TN-SS problem without any priors on the domain information. Without domain information, the LLM generates a poorly initialized TN structure that is 80.3% worse in terms of objective function value, due to its lack of information about the different modes, and requires $10.4\times$ more evaluations to converge. Moreover, as the model defaults to random assumptions about the modes, as illustrated in Figure 6, it was found to produce uniformly connected TN structures throughout the minimization process. This restricts the search space to a much smaller set of solutions that are not domain-meaningful from a practical perspective.

**Selection of LLM models.** To examine how the choice of LLM affects the performance of tnLLM, we compared our baseline model, GPT-4o (`gpt-4o-2024-08-06`), against GPT-4.5 (`gpt-4.5-preview-2025-02-27`), GPT4o-mini (`gpt-4o-mini-2024-07-18`), GPT-3.5 (`gpt-3.5-turbo-1106`) and the open-source DeepSeek V3 (`DeepSeek-V3-0324`) model. Observe from Table 3 that due to the carefully designed prompts and the structure of the overall framework, tnLLM is robust to the choice of the LLM used. While there are small variations in the achieved objective function values, the overall performance remained consistent across all three datasets in both training and testing sets. Furthermore, the number of evaluations required by all LLMs falls within the mean ± standard deviation range reported in Table 1. It is important to note that, despite the consistent performance, weaker models and in particular GPT-3.5 occasionally misinterpreted the optimization objective, mistakenly assuming that higher values were preferable.

Table 3: Ablation study of using different LLMs within the proposed framework. For different LLMs, we report the best training objective values achieved and their corresponding testing objective values.

| Dataset | GPT-4o (Baseline) | | GPT-4.5 | | GPT-4o-mini | | GPT-3.5 | | DeepSeek-V3 | |
|---|---|---|---|---|---|---|---|---|---|---|
| | Train | Test | Train | Test | Train | Test | Train | Test | Train | Test |
| Images | -0.63 | -0.48 | -0.62 | -0.54 | -0.61 | -0.48 | -0.64 | -0.52 | -0.62 | -0.47 |
| Videos | -1.63 | -1.70 | -1.62 | -1.65 | -1.62 | -1.69 | -1.63 | -1.70 | -1.63 | -1.70 |
| Time-series | -0.42 | -0.41 | -0.40 | -0.41 | -0.41 | -0.40 | -0.42 | -0.40 | -0.40 | -0.39 |

## 5 CONCLUSION

We have introduced tnLLM, a domain-aware LLM-guided TN-SS framework for directly infering TN structures. This has been achieved by utilizing the rich domain information in real-world tensor data and the inherent reasoning capabilities of LLMs. Experimental results have demonstrated that tnLLM achieves performance comparable to current SOTA algorithms, while requiring significantly fewer function evaluations. Notably, by incorporating domain information, our framework is the first to mitigate the black-box nature of the identified TN structures in TN-SS through generating domain-relevant solution explanations. Furthermore, we have shown that tnLLM can be used to accelerate SOTA sampling-based algorithms while preserving their theoretical guarantees.

## REPRODUCIBILITY STATEMENT

Details of the overall framework and prompts used are provided in Section 3 and Appendix B. Experimental settings are provided in Section 4. Implementation details of the baseline models are provided in Appendix D. Details of the data are provided in Section 4 and Appendix C.

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

## A    LIMITATIONS AND FUTURE WORK

Despite the good performance, the proposed domain-aware TN-SS framework currently lacks theoretical guarantees for its evaluation efficiency due to the black-box nature of LLMs. This could be mitigated by developing domain-aware TN-SS algorithms based on continuous optimization. Also, as mentioned in the ablation study, weaker LLM models, such as GPT-3.5, were found to be more prone to misinterpret the optimization objectives. Therefore, improving the reasoning consistency of smaller LLM models promises to further improve the efficiency of the framework. Furthermore, prompt learning is an active area of research in LLMs, which can be used to potentially enhance the tnLLM framework in future works.

## B    FULL PROMPTS OF TNLLM FRAMEWORK

The full "Behavior-directive", "Task-directive" and "Optimization-directive" prompts used in the tnLLM framework are illustrated in Figures 8, 9, 10, respectively.

> """ You are a <domain> expert specialized in tensor decomposition. Your task is to <task description>. You need to provide <solution description> which minimizes the loss function, which is the natural log of the sum of the compression rate and 10 times the approximation error. The compression ratio is calculated as the number of parameters in the compressed FCTND format divided by the original number of parameters of the uncompressed tensor and the approximation error is the relative square error between the original and approximate tensor. Work your suggestions out step-by-step based on rigorous reasoning and <domain> knowledge. Explain your final suggestions in a logical, concise manner."""

Figure 8: The full "Behavior-directive" prompt.

> """We are working with a fully connected <tensor size> tensor representing <domain> data with the following modes:
> <Description of each mode including its size and content>
>
> Your task is to suggest the optimal solution for each connection in a fully connected tensor network decomposition. The loss function to minimize is a natural log of the sum of the compression rate and 10 times the tensor approximation error which is the relative square error between the original and approximate tensor. Provide your response in the following format:
> 1. Take a deep breath and reason step-by-step about the intrinsic interactions between every pair of modes based on your understanding of the relationships about <mode information>. It is important to reason about those intrinsic interactions based on interpretable factors.
> 2. Based on your reasoning, output <solution desciption>.
> Output format:
> Reasoning: Reason about the intrinsic interactions between every pair of modes based on your understanding of <domain> data.
>
> Solution: <Exact solution format>
> End the output message with <description of solution format>. """

Figure 9: The full "Task-directive" prompt.

## C    DATA

We constructed a custom financial time series dataset with 142 temporally ordered fifth-order tensors, denoted as $\{\mathcal{X}_n\}_{n=1}^{142} \in \mathbb{R}^{3 \times 6 \times 3 \times 4 \times 5}$, each representing a rolling window produced via multi-way delay embedding through the temporal direction (Yokota et al., 2018). This leads to the value selection process of 10 ranks. To the best of our knowledge, this is the largest tensor dataset in terms of number of samples ever considered in the TN-SS problem. The first $80\%$ of these tensors were used as the training data, while the remaining $20\%$ with non-overlapping entries with the training data were used for testing. The modes of each time series tensor correspond to:

- **Mode 1**: Types of financial instruments. They are equity indices, commodities, and currency swaps.

""" The last solution is <last_solution> with a total loss function of <last loss>, which is the natural log of the sum of the current compression rate of <last_compression_rate> and 10 times the current approximation error of <last_approximation_error>. The lowest total loss function is <best_objective_function}> which is the natural log of the sum of the compression rate of <best_compression_rate> and ten times the approximation error of <best_approximation_error}> is found by using the solution <best_solution>, The loss function to minimize is a natural log of the sum of the compression rate and 10 times the tensor approximation error which is the relative square error between the original and approximate tensor.

Take a deep breath, refine the solution suggestions to make the loss function smaller, and justify any changes in the solution. Keep in mind that increasing the ranks significantly decreses the approximation error, while it increases the compression rate. However, if the compression rate is already very low compared to the approximation error (for example, the compression rate is smaller than half of the approximation error), increasing the ranks (such as doubling it) to decrease the approximation error will usually lead to a lower loss function. Also, if the compression rate is already very high compared to the approximation error (for example, the compression rate is larger than two times the approximation error), reducing the ranks (such as cutting it in half) to decrease the compression rate will usually lead to a lower loss function.

When refining the ranks, consider how each mode <mode titles> interacts with the others and how reducing or increasing the rank will affect the overall decomposition accuracy. You are encouraged to be explorative to try small and large rank value changes in this process. Do not change ranks if you think its not necessary. You should never try the same set of ranks more than once. Trying the same set of ranks more than once wastes computation resources and will not lead to a different outcome.

Provide the solution and reason step-by-step for the changes in the following format:
Output format:
Reasoning: Reason about the intrinsic interactions between every pair of modes based on your understanding of <domain> data.

Solution: <Exact solution format>
End the output message with <description of solution format>. """

Figure 10: The full "Optimization-directive" prompt.

- **Mode 2**: Assets within each type of financial instrument. For equity indices, these are Hang Seng, Nikkei 225, S&P 500, EURO STOXX 50, FTSE 100, and Shanghai Composite Index. For commodities, these are Brent Crude, Copper, Natural gas, Comex gold, Soybeans, and Wheat. For currency swaps, these are HKD/USD, JPY/USD, CHF/USD, EUR/USD, CNY/USD, and GBP/USD.
- **Mode 3**: Features of each asset. These are average adjusted closing price log return, average relative price min-max, and average high-low spread.
- **Mode 4**: Interval of time points on which we calculate the average features. There are 4 intervals – 1 day, 5 days, 10 days, and 15 days.
- **Mode 5**: Time points within each rolling window of length 5.

The images and videos datasets are fetched from *http://trace.eas.asu.edu/yuv/*. Figure 11 illustrates 5 example image samples, $\{\mathcal{X}_n\}_{n=1}^5 \in \mathbb{R}^{144 \times 176 \times 3}$, used in the experiments. The first $80\%$ of these tensors were used as the training data, while the remaining $20\%$ for testing. The modes of each image tensor correspond to:

- **Mode 1**: Height of the image. These are the vertical pixel indexes.
- **Mode 2**: Width of the image. These are the horizontal pixel indexes.
- **Mode 3**: RGB channels. These are the red, green, and blue channels.

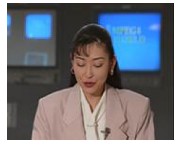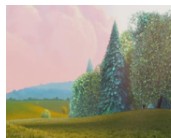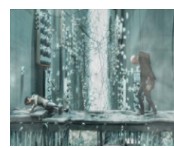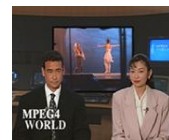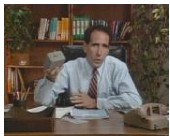

Figure 11: Image samples used in the experiments.

Video dataset is produced from performing the multi-way delay embedding (Yamamoto et al., 2022) through the temporal mode in a video to produce 5 samples, $\{\mathcal{X}_n\}_{n=1}^5 \in \mathbb{R}^{144 \times 176 \times 3 \times 10}$. Figure 12

shows an example sample of the videos dataset at different frames. The modes of each video tensor correspond to:

- **Mode 1**: Height of the image. These are the vertical pixel indexes.
- **Mode 2**: Width of the image. These are the horizontal pixel indexes.
- **Mode 3**: RGB channels. These are the red, green, and blue channels.
- **Mode 4**: Frames in a video. These are the index of the frames in a video.

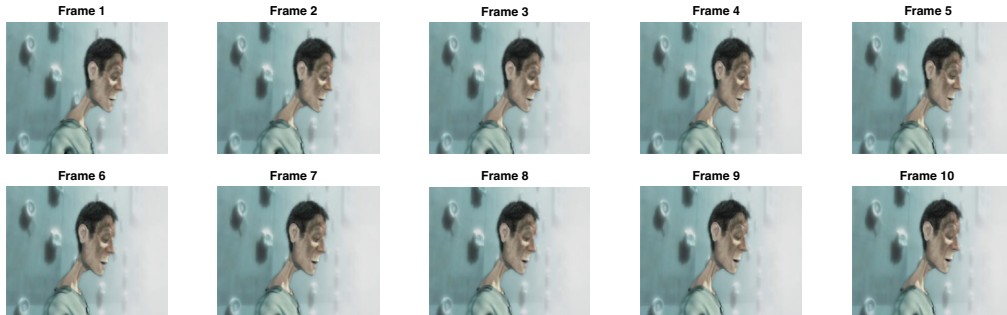

Figure 12: One example sample of the Video data used in the experiments.

## D    IMPLEMENTATION DETAILS

All baseline models are initialized as a "fully-disconnected" graph. The TN search template is set to a complete graph. For TNLS, we set the number of samples (evaluations), $\#Sample$, in each sampling step to 4 in the images dataset, 60 in the videos dataset, and 10 in the time series dataset. For TnALE, we set $D = 1$, $r = 1$. For tnGPS, we selected the best-performing algorithm discovered ("Ho-2") and adopted the following hyperparameters as suggested in their paper: code upper bound of 10, mutation rate of 0.1, crossover rate of 0.6, selection pressure of 1.5, elitism enabled, diversity factor of 0.05, variance decay of 0.98, minimum variance of 0.1, tournament size factor of 0.2, elite diversity boost of 2.0, random individual chance of 0.05, and a maximum mutation of 3. All models were run for 500 evaluations in both the images and videos dataset, and 250 evaluations in the time series dataset if not converged. An internal server with NVIDIA RTX A6000 GPU, an AMD Ryzen Threadripper PRO 5955WX with 16 cores, and 256GB of RAM was used.

## E    RUNTIME COMPARISONS

Table 4 reports the total average runtime in seconds to first achieve the best training objective function value for all models across the three datasets of Table 1. For tnLLM, this also includes the LLM inference time. The proposed tnLLM achieves comparable performance to SOTA methods while reducing runtime by up to 98.3% compared to TNLS, 97.7% compared to TnALE, and 98.1% compared to tnGPS.

Table 4: Runtime comparisons across three datasets. The values are the total average runtime in seconds to first achieve the best training objective function value. A lower value is better. The best values are denoted in bold. The second best values are underlined.

| Data Type | TNLS | TnALE | tnGPS | tnLLM (Ours) |
|---|---|---|---|---|
| Images | 2,122s | 1,659s | 8,025s | **151s** |
| Videos | 17,690s | 15,298s | 20,461s | **1,984s** |
| Time-series | 110,945s | 80,700s | 17,287s | **1,866s** |

# F    SUMMARIZED GENERATED EXPLANATIONS OF TNLLM FOR THE INITIALIZED TN STRUCTURE

The full set of summarized LLM generated explanations over three independent runs for the images, videos, and time-series datasets are provided in Tables 5, 6, and 7, respectively. The suggested ranks are verified to be consistent with the domain-informed reasoning provided by tnLLM. Also, the domain-relevant explanations are robust across multiple runs.

Table 5: Summarized generated explanations of tnLLM for the initialized TN structure in the Images dataset over 3 independent runs.

| Mode Pair | Run 1 | Run 2 | Run 3 |
|---|---|---|---|
| **Mode 1 & Mode 2** (Width ↔ Height) | • Defines spatial resolution
• Spatial coherence is significant due to smoothness
• **High rank** needed to capture spatial detail (suggested rank: 20) | • Spatial dimensions are highly correlated
• Captures patterns and structures
• **High rank** to capture spatial complexity (suggested rank: 20) | • Spatial resolution key for details and patterns
• **High rank** crucial to capture more spatial features to maintain image quality
• **High rank** (suggested rank: 20) |
| **Mode 1 & Mode 3** (Width ↔ RGB) | • Captures horizontal color distribution
• Less complex than spatial features
• **Moderate rank** to capture color variation without overfitting (suggested rank: 5) | • Horizontal color changes are smooth
• Color variations less complex as object maintains consistent color across width
• **Moderate rank** (suggested rank: 5) | • Related through color distribution along width
• Color variations less complex than spatial variation
• **Moderate rank** (suggested rank: 5) |
| **Mode 2 & Mode 3** (Height ↔ RGB) | • Captures vertical color patterns
• Vertical and horizontal color distributions often equally important
• Rank should be **similar** to the width–RGB connection (suggested rank: 5) | • Captures vertical color variation
• Vertical structures (e.g., stripes) may add complexity
• Rank might be **slightly higher** than width–channel rank (suggested rank: 8) | • Complexity of color variation along height and width typically similar
• **Moderate rank** is also appropriate (suggested rank: 5) |

# G    PERFORMANCE COMPARISON FOR VANILLA AND TNLLM-INITIALIZED TNLS AND TNALE

Table 8 illustrates the detailed performance of vanilla and tnLLM-initialized TNLS and TnALE across different datasets.

Table 6: Summarized generated explanations of tnLLM for the initialized TN structure in the Videos dataset over 3 independent runs.

| Mode Pair | Run 1 | Run 2 | Run 3 |
|---|---|---|---|
| **Mode 1 & Mode 2**
(Width ↔ Height) | • Defines spatial resolution per frame
• Needs to capture spatial detail for video quality but balance with compression
• **Moderate-to-high rank** (suggested rank: 20) | • Define spatial resolution per frame
• Higher rank captures more complex spatial patterns and correlations
• **Moderate-to-high rank** to balance detail capture and compression (suggested rank: 20) | • Define spatial resolution per frame
• Spatial resolution is a significant aspect of video quality
• **Moderate-to-high rank** to capture more spatial details across frames (suggested rank: 20) |
| **Mode 1 & Mode 3, Mode 2 & Mode 3**
(Width/Height ↔ RGB) | • Less complex than spatial info
• Limited variation of color across width/height
• **Low rank** suffices (suggested rank: 5) | • Less directly correlated than spatial dimensions
• Certain horizontal/vertical patterns prominent in specific color channels
• **Low rank** suffices (suggested rank: 5) | • Captures color variations horizontally/vertically
• Color variations can be significant in videos with rich color content
• **Moderate-to-low rank** (suggested rank: 5) |
| **Mode 1 & Mode 4, Mode 2 & Mode 4**
(Width/Height ↔ Frames) | • Captures horizontal/vertical structure changes over time
• Videos have temporal consistency
• **Moderate rank** (suggested rank: 10) | • Related through motion and changes across frames
• **Moderate rank** to capture temporal changes across width/height of the video (suggested rank: 10) | • Captures temporal variations across the horizontal/vertical dimension
• Temporal changes crucial for motion representation
• **Moderate rank** (suggested rank: 10) |
| **Mode 3 & Mode 4**
(RGB ↔ Frames) | • Captures color changes over time
• Color changes often subtle and less complex than spatial changes
• **Lower rank** can be used to maintain compression (suggested rank: 3) | • Captures color changes over time
• Crucial for capturing dynamic color variations and transitions
• **Low rank** to model temporal color variation (suggested rank: 5) | • Captures temporal color changes
• Essential for representing dynamic scenes
• **Moderate-to-low rank** necessary to capture these changes (suggested rank: 5) |

Table 7: Summarized generated explanations of tnLLM for the initialized TN structure in the Time-series dataset over 3 independent runs.

| Mode Pair | Run 1 | Run 2 | Run 3 |
|---|---|---|---|
| **Mode 1 & Mode 2** (Types of financial instruments ↔ Assets within each type) | • Strong intrinsic relationship
• Different assets within a type exhibit correlated behaviors due to shared market factors
• **Higher rank** (suggested rank: 3) | • Strong relationship as each type is defined by its constituent assets
• High diversity and specificity across assets
• **Higher rank** (suggested rank: 3) | • Strong connection as each instrument has specific assets
• Rank should reflect the diversity and specificity of assets within each type
• **Higher rank** (suggested rank: 3) |
| **Mode 1 & Mode 3** (Types of financial instruments ↔ Features of each asset) | • Features influenced by instrument type
• For example, volatility of commodities might differ from that of equity indexes
• **Moderate rank** (suggested rank: 2) | • Different instruments behave distinctly (returns, volatility, spreads)
• Features somewhat standardized across types
• **Moderate rank** to not overfit (suggested rank: 2) | • Instruments behave differently in terms of features
• However, features themselves are common across all types
• **Moderate rank** to capture variability (suggested rank: 2) |
| **Mode 1 & Mode 4** (Types of financial instruments ↔ Interval of time points) | • Different financial instruments react differently over time intervals due to inherent market dynamics
• **Moderate rank** (suggested rank: 2) | • Financial instruments differ in how sensitive they are to short and long-term time intervals
• **Moderate rank** (suggested rank: 2) | • Impact of time intervals on financial instruments can vary.
• Commodities react differently to short-term versus long-term intervals compared to equity indexes.
• **Moderate rank** (suggested rank: 2) |
| **Mode 1 & Mode 5** (Types of financial instruments ↔ Time points in each rolling window) | • Reaction of financial instruments to specific time points can vary
• This interaction might not be as strong as others
• **Lower** rank (suggested rank: 1) | • Financial instruments show low sensitivity to individual time points
• Rolling windows naturally smooth out short-term fluctuations
• **Lower** rank (suggested rank: 1) | • Specific time points affect different instruments differently
• This interaction is likely less significant than others
• **Lower** rank (suggested rank: 1) |
| **Mode 2 & Mode 3** (Assets within each type ↔ Features of each asset) | • Features directly derived from assets indicating strong relationship
• Different assets have distinct feature profiles
• **Higher rank** (suggested rank: 3) | • Each asset has unique characteristics in terms of returns, volatility, and spreads
• **Higher rank** to accurately capture specific features (suggested rank: 3) | • Each asset has unique characteristics in terms of features like log returns and spreads
• **Higher rank** needed for this detailed interaction (suggested rank: 3) |
| **Mode 2 & Mode 4** (Assets within each type ↔ Interval of time points) | • Performance of assets over different time intervals can vary significantly, especially in volatile markets.
• **Moderate** rank (suggested rank: 2) | • Assets respond differently to various time intervals
• Some assets exhibit stronger trends or higher volatility over specific periods
• **Moderate** rank (suggested rank: 2) | • Different assets exhibit different behaviors over various time intervals.
• Some assets may be more volatile in the short term.
• **Moderate** rank to capture these dynamics (suggested rank: 2) |

| **Mode 2 & Mode 5** (Assets within each type ↔ Time points in each rolling window) | • Specific time points within a rolling window can affect asset performance
• This interaction might be less pronounced
• **Lower** rank (suggested rank:1) | • Influence of time points on asset features is minimal
• Rolling window averages out short-term noise.
• **Lower** rank (suggested rank:1) | • Specific time points within a rolling window can affect asset performance
• This interaction is likely less significant than others
• **Lower** rank (suggested rank:1) |
|---|---|---|---|
| **Mode 3 & Mode 4** (Features of each asset ↔ Interval of time points) | • Features such as log returns and spreads are sensitive to the time interval chosen.
• **Moderate** rank to capture temporal effects (suggested rank: 2) | • Features such as returns and volatility can vary significantly over different time intervals.
• **Moderate** rank (suggested rank: 2) | • Features like log returns and spreads can vary significantly over different time intervals.
• Connection is strong as the features are calculated over these intervals.
• Higher rank (suggested rank: 3) |
| **Mode 3 & Mode 5** (Features of each asset ↔ Time points in each rolling window) | • Features can vary with specific time points within a rolling window, but interaction is weaker
• **Lower rank** is suitable (suggested rank: 1) | • Specific time points have a limited impact on features due to averaging effects.
• **Lower rank** is suitable (suggested rank: 1) | • Specific time points within a rolling window can affect the calculation of features
• This interaction is likely less significant than others.
• **Lower rank** is suitable (suggested rank: 1) |
| **Mode 4 & Mode 5** (Interval of time points ↔ Time points in each rolling window) | • Inherently linked as rolling window defines the context for the time points
• **Moderate rank** (suggested rank: 2) | • Weak relationship as rolling window smooths out short-term variations
• **Lower rank** is suitable (suggested rank: 1) | • Intrinsic connection as time points are nested within the intervals
• **Moderate rank** to capture this structure (suggested rank: 2) |

Table 8: Performance comparison across different datasets for vanilla and tnLLM-initialized TNLS and TnALE. The values on the left give the lowest training objective function values and their corresponding testing objective function values. The values in [square brackets] give the number of evaluations required to first achieve the best training objective function value. For both metrics, a lower value is better. The least number of evaluations are highlighted in bold.

| Data Type | | TNLS | | | | TnALE | | | |
|---|---|---|---|---|---|---|---|---|---|
| | | Vanilla | | tnLLM-init | | Vanilla | | tnLLM-init | |
| **Images** | Train
Test | -0.66
-0.47 | [114] | -0.64
-0.46 | [**20**] | -0.65
-0.46 | [81] | -0.64
-0.46 | [**19**] |
| **Videos** | Train
Test | -1.64
-1.72 | [484] | -1.64
-1.71 | [**21**] | -1.66
-1.72 | [254] | -1.65
-1.70 | [**20**] |
| **Time-series** | Train
Test | -0.45
-0.43 | [218] | -0.47
-0.44 | [**14**] | -0.47
-0.47 | [177] | -0.47
-0.44 | [**20**] |

# H    PERFORMANCE COMPARISON WITH EARLY STOPPING

To further assess the speed-up benefits of the proposed tnLLM method, we incorporate an early stopping criterion for the baseline methods, which is not present in their original implementations. Specifically, Table 9 reports the convergence behavior of all baselines using an early stopping threshold of 0.01 on the objective function, with a patience of 15, applied across all three datasets. observe from Table 9 that even with early stopping the proposed tnLLM is able to achieve significant speed ups in number of evaluations (up to 68x compared to TNLS, 24x compared to TnALE, and 53x compared to tnGPS), while the objective values achieved by the baselines are now slightly worse relative to those reported in Table 1.

Table 9: Performance comparison across different datasets. The values on the left give the lowest training and corresponding testing objective function values. The values in [square brackets] give the number of evaluations required to first achieve the best training objective function value. For all methods, we report the average and standard deviation of both the objective function value and the number of evaluations across 5 independent runs. For both metrics, a lower value is better. The best values are denoted in bold. The second best values are underlined.

| Data Type | | TNLS | | TnALE | | tnGPS | | tnLLM (Ours) | |
|---|---|---|---|---|---|---|---|---|---|
| Images | Train | $-0.64 \pm 0.01$ | $[58.6 \pm 5.2]$ | $-0.63 \pm 0.02$ | $[\underline{51.2} \pm 10.4]$ | $-0.64 \pm 0.01$ | $[210.0 \pm 22.3]$ | $-0.63 \pm 0.01$ | $[\mathbf{4.0} \pm 1.9]$ |
| | Test | $\underline{-0.46} \pm 0.02$ | | $\underline{-0.46} \pm 0.01$ | | $-0.42 \pm 0.03$ | | $\mathbf{-0.48} \pm 0.05$ | |
| Videos | Train | $-1.63 \pm 0.02$ | $[422.2 \pm 20.1]$ | $-1.66 \pm 0.01$ | $[\underline{148.8} \pm 11.5]$ | $-1.65 \pm 0.01$ | $[185.0 \pm 26.8]$ | $-1.63 \pm 0.01$ | $[\mathbf{6.2} \pm 3.5]$ |
| | Test | $-1.70 \pm 0.03$ | | $\mathbf{-1.72} \pm 0.01$ | | $\underline{-1.71} \pm 0.02$ | | $-1.70 \pm 0.02$ | |
| Time-series | Train | $-0.46 \pm 0.01$ | $[170.6 \pm 7.6]$ | $-0.46 \pm 0.03$ | $[115.2 \pm 14.4]$ | $-0.40 \pm 0.02$ | $[\underline{62.6} \pm 17.0]$ | $-0.42 \pm 0.02$ | $[\mathbf{5.6} \pm 4.4]$ |
| | Test | $\underline{-0.42} \pm 0.01$ | | $\mathbf{-0.44} \pm 0.02$ | | $-0.40 \pm 0.02$ | | $-0.41 \pm 0.02$ | |

# I    EXPLANATIONS OF TNLLM DURING TN-DISCOVERY STAGE

The full outputs of the proposed tnLLM during the TN-discovery stage for a single run on the images, video and time-series datasets are provided below.

**Images Dataset**

---

**TN Initialization Rank Array**: [20, 5, 5]
**Loss Function at iteration 0**: -0.5737094673

---

**TN Discovery**

**Running iteration 1 of iterative prompt**:
**Detailed Reasoning**: To refine the rank suggestions for the Fully Connected Tensor Network Decomposition (FCTND) of an RGB image tensor, we need to balance the trade-off between the compression rate and the approximation error. The goal is to minimize the loss function, which is the natural log of the sum of the compression rate and ten times the approximation error.

**1. Current Situation Analysis**:
- The current rank array is [20, 5, 5] with a loss function of -0.5737094673.
- The compression rate is 0.4218618476, and the approximation error is 0.141569679.
**2. Understanding the Trade-offs**:
- Compression Rate: This is the ratio of parameters in the compressed tensor to the original tensor. A lower compression rate means more compression.
- Approximation Error: This is the relative square error between the original and approximate tensor. A lower approximation error indicates a better approximation.
**3. Guidelines for Adjusting Ranks**:
- If the compression rate is much lower than half of the approximation error, increasing the ranks might reduce the approximation error significantly, potentially lowering the loss function.

- If the compression rate is much higher than twice the approximation error, reducing the ranks might lower the compression rate, potentially lowering the loss function.

**4. Current Comparison**:
- The compression rate (0.4218618476) is approximately three times the approximation error (0.141569679). This suggests that reducing the ranks could be beneficial, as the compression rate is relatively high compared to the approximation error.

**5. Rank Adjustment Strategy**:
- Since the compression rate is significantly higher than twice the approximation error, we should consider reducing the ranks to decrease the compression rate.
- We should explore reducing the rank for the connection between the width and height (Mode 1 and Mode 2), as this is typically the largest dimension in an image tensor and might have the most significant impact on the compression rate.

**6. Proposed Rank Changes**:
- Reduce the rank for the connection between Mode 1 (width) and Mode 2 (height) from 20 to 10. This is a moderate reduction that should help decrease the compression rate without drastically increasing the approximation error.
- Keep the ranks for the connections between Mode 1 (width) and Mode 3 (RGB channels), and Mode 2 (height) and Mode 3 (RGB channels) at 5, as these are already relatively low and further reduction might significantly increase the approximation error.

**Rank Array**: [10, 5, 5]
This proposed rank array aims to balance the compression rate and approximation error more effectively, potentially leading to a lower loss function.

**Loss Function at iteration 1**: -0.5521177879

---

**Running iteration 2 of iterative prompt**:
**Detailed Reasoning**: Let's analyze the situation with the given rank arrays and their corresponding loss functions. We have two sets of ranks and their associated loss functions:

1. **Last Rank Array** [10, 5, 5]:
- Compression Rate: 0.2114241372
- Approximation Error: 0.3643051088
- Loss Function: -0.5521177879

2. **Best Rank Array** [20, 5, 5]:
- Compression Rate: 0.4218618476
- Approximation Error: 0.141569679
- Loss Function: -0.5737094673

The goal is to minimize the loss function, which is the natural log of the sum of the compression rate and ten times the approximation error.

**Analysis**:
- **Compression Rate vs. Approximation Error**:
- The rank array [20, 5, 5] has a higher compression rate but a significantly lower approximation error compared to [10, 5, 5]. This results in a lower loss function, indicating that the approximation error has a more substantial impact on the loss function than the compression rate in this scenario.
- The compression rate for [10, 5, 5] is much lower than the approximation error, suggesting that increasing ranks could potentially lower the loss function by reducing the approximation error.
- **Intrinsic Interactions**:
- In RGB images, the interaction between the width and height (spatial dimensions) is crucial for maintaining image structure. The interaction between spatial dimensions and the RGB channels is also important for color fidelity.
- Increasing the rank between the spatial dimensions (Mode 1 and Mode 2) could significantly reduce the approximation error, as it allows for better representation of spatial patterns.
- The rank between the spatial dimensions and the RGB channels (Mode 1 and Mode 3, Mode 2 and Mode 3) should be balanced to maintain color accuracy without excessively increasing the compression rate.

**Proposed Changes**:

Given the analysis, we should explore increasing the rank between the spatial dimensions (Mode 1 and Mode 2) to reduce the approximation error, while keeping the ranks involving the RGB channels moderate to avoid a high compression rate.
- Rank for (Mode 1, Mode 2): Increase from 10 to 15 to improve spatial representation.
- Rank for (Mode 1, Mode 3) and (Mode 2, Mode 3): Maintain at 5 to balance color accuracy and compression rate.
**Rank Array**: [15, 5, 5]
This adjustment aims to reduce the approximation error significantly while keeping the compression rate in check, potentially leading to a lower loss function.
**Loss Function at iteration 2**: -0.6382834296

### Videos Dataset

**TN Initialization**
**Rank Array**: [20, 5, 10, 5, 10, 3]
**Loss Function at iteration 0**: -0.8044418714

---

**TN Discovery**

**Running iteration 1 of iterative prompt**:
**Detailed Reasoning**: To refine the rank suggestions and minimize the loss function, we need to carefully balance the compression rate and the approximation error. The current rank array is [20, 5, 10, 5, 10, 3], with a compression rate of 0.425117056 and an approximation error of 0.0222204728. The loss function is calculated as the natural log of the sum of the compression rate and ten times the approximation error.
**1. Compression Rate vs. Approximation Error**: The current compression rate (0.425117056) is significantly larger than the approximation error (0.0222204728). This suggests that we might benefit from reducing the ranks to decrease the compression rate, as the compression rate is more than two times the approximation error.
**2. Mode Interactions**:
- Width and Height: These are spatial dimensions and typically have strong correlations. Reducing the rank between these modes might slightly increase the approximation error but could significantly reduce the compression rate.
- RGB Channels: The interaction between RGB channels and other modes (width, height, frames) is crucial for maintaining color fidelity. However, the rank between RGB channels and other modes can often be reduced without a significant impact on the approximation error.
- Frames: Temporal coherence is important, but reducing the rank between frames and spatial dimensions can often be done with minimal impact on the approximation error.
**3. Rank Adjustments**:
- **(Mode 1, Mode 2)**: Reduce from 20 to 15. This should decrease the compression rate significantly while maintaining a reasonable approximation error.
- **(Mode 1, Mode 3)**: Reduce from 5 to 4. This small reduction should help with the compression rate without a large impact on error.
- **(Mode 1, Mode 4)**: Reduce from 10 to 8. This balances the spatial and temporal dimensions.
- **(Mode 2, Mode 3)**: Reduce from 5 to 4. This should help reduce the compression rate.
- **(Mode 2, Mode 4)**: Reduce from 10 to 8. This maintains a balance between spatial and temporal coherence.
- **(Mode 3, Mode 4)**: Reduce from 3 to 2. This is a conservative reduction to help with compression.
By making these adjustments, we aim to reduce the compression rate more significantly than the increase in approximation error, thereby minimizing the loss function.
**Rank Array**: [15, 4, 8, 4, 8, 2]
**Loss Function at iteration 1**: -1.3477094611

---

**Running iteration 2 of iterative prompt**:

**Detailed Reasoning**: To refine the rank suggestions, we need to balance the compression rate and approximation error to minimize the loss function, which is the natural log of the sum of the compression rate and ten times the approximation error. The current rank array is [15, 4, 8, 4, 8, 2], with a compression rate of 0.2038299663 and an approximation error of 0.0560047748.

**1. Compression Rate vs. Approximation Error**: The compression rate (0.2038) is approximately 3.64 times the approximation error (0.0560). This suggests that the compression rate is relatively high compared to the approximation error. According to the guideline, if the compression rate is larger than two times the approximation error, reducing the ranks to decrease the compression rate might lead to a lower loss function.

**2. Mode Interactions**: In RGB video tensors, the modes typically represent width, height, RGB channels, and frames. The interaction between these modes can be complex, but generally:
- The connection between spatial dimensions (width and height) often requires higher ranks to capture spatial patterns.
- The connection between RGB channels might not need as high ranks since color information is often correlated.
- Temporal connections (frames) can vary depending on the video content, but often require moderate ranks to capture motion.

**3. Adjusting Ranks:**
- (**Mode 1, Mode 2) and (Mode 2, Mode 3**): These connections involve spatial dimensions and RGB channels. Given the high compression rate, we can try reducing these ranks slightly to see if we can reduce the compression rate without significantly increasing the approximation error.
- (**Mode 1, Mode 3) and (Mode 2, Mode 4**): These involve spatial and temporal dimensions. We should be cautious in reducing these too much as they are crucial for capturing spatial-temporal patterns.
- (**Mode 3, Mode 4**): This involves RGB channels and frames. We can try reducing this rank slightly as RGB channels often have correlated information.

Based on these considerations, let's propose a new rank array:

**Rank Array**: [12, 3, 8, 3, 7, 2]
- (**Mode 1, Mode 2**): Reduced from 15 to 12 to decrease the compression rate.
- (**Mode 1, Mode 3**): Reduced from 4 to 3 to slightly decrease the compression rate.
- (**Mode 1, Mode 4**): Kept at 8 to maintain spatial-temporal accuracy.
- (**Mode 2, Mode 3**): Reduced from 4 to 3 to slightly decrease the compression rate.
- (**Mode 2, Mode 4**): Reduced from 8 to 7 to slightly decrease the compression rate while maintaining temporal accuracy.
- (**Mode 3, Mode 4**): Kept at 2 as RGB and temporal interaction might not need high ranks.

This adjustment aims to reduce the compression rate while keeping the approximation error manageable, potentially leading to a lower loss function.

**Current rank array is**: [12, 3, 8, 3, 7, 2]

**Loss Function at iteration 2**: -1.5880439406

---

**Running iteration 3 of iterative prompt**:

**Detailed Reasoning**: Let's start by analyzing the current situation. The current rank array is [12, 3, 8, 3, 7, 2], with a compression rate of 0.1144228746 and an approximation error of 0.0899020181. The loss function is calculated as the natural log of the sum of the compression rate and ten times the approximation error, resulting in a value of -1.5880439406.

**1. Compression Rate vs. Approximation Error**: - The compression rate (0.114) is slightly higher than the approximation error (0.0899). This suggests that the compression rate is not excessively high compared to the approximation error, but it is not significantly lower either. Therefore, we should aim to balance these two factors more effectively.

**2. Intrinsic Interactions**:
- In RGB video data, the interactions between modes such as width, height, RGB channels, and frames are crucial. The RGB channels often have less variability compared to spatial dimensions (width and height) and temporal dimensions (frames). Therefore, we can consider adjusting ranks related to RGB channels more conservatively.
- The spatial dimensions (width and height) are typically more correlated, and increasing ranks here could potentially reduce the approximation error significantly.

**3. Explorative Adjustments**:
- Given the current ranks, we can explore increasing the ranks where the approximation error might be reduced more effectively. Since the compression rate is not excessively low, we can afford to increase some ranks to see if the approximation error decreases more than the increase in the compression rate.
- We should consider increasing the ranks between spatial dimensions (width and height) and temporal dimensions (frames) as these are likely to have more complex interactions.

**4. Proposed Changes**:
- Increase the rank between width and height (Mode 1 and Mode 2) from 12 to 16. This is because spatial dimensions often have complex interactions that can benefit from higher ranks.
- Increase the rank between width and frames (Mode 1 and Mode 4) from 8 to 10. This might help capture more temporal dynamics.
- Increase the rank between height and frames (Mode 2 and Mode 4) from 7 to 9. This will further help in capturing temporal dynamics.

**5. Maintaining Other Ranks**:
- Keep the rank between width and RGB channels (Mode 1 and Mode 3) at 3, as RGB channels typically do not require high ranks.
- Keep the rank between height and RGB channels (Mode 2 and Mode 3) at 3 for the same reason.
- Keep the rank between RGB channels and frames (Mode 3 and Mode 4) at 2, as this interaction is typically less complex.

By making these adjustments, we aim to reduce the approximation error more significantly than the increase in the compression rate, potentially leading to a lower loss function.

**Rank Array**: [16, 3, 10, 3, 9, 2]

**Loss Function at iteration 3**: -1.3734126669

---

**Running iteration 4 of iterative prompt**:

**Detailed Reasoning**:

To refine the rank suggestions, we need to balance the compression rate and the approximation error to minimize the loss function. The loss function is the natural log of the sum of the compression rate and ten times the approximation error. Our goal is to find a rank configuration that results in a lower loss function than the previous best of -1.5880439406.

**1. Understanding the Current State**:
- The last rank array [16, 3, 10, 3, 9, 2] resulted in a loss function of -1.3734126669.
- The best rank array [12, 3, 8, 3, 7, 2] resulted in a loss function of -1.5880439406.

**2. Analyzing Compression Rate and Approximation Error**:
- The compression rate for the best configuration was 0.1144228746, and the approximation error was 0.0899020181.
- The compression rate for the last configuration was 0.1933475379, and the approximation error was 0.0598937173.

**3. Strategy for Refinement**:
- Since the compression rate in the best configuration is lower than half of the approximation error, increasing the ranks could potentially decrease the approximation error significantly, leading to a lower loss function.
- Conversely, if the compression rate is high relative to the approximation error, reducing ranks could be beneficial.

**4. Explorative Adjustments**:
- **Mode 1 and Mode 3 (Width and RGB Channels)**: The rank was 3 in both configurations. Keeping it constant might be beneficial as it balances compression and error.
- **Mode 1 and Mode 4 (Width and Frames)**: The rank was 10 in the last configuration and 8 in the best configuration. Increasing this rank could reduce the approximation error significantly.
- **Mode 2 and Mode 3 (Height and RGB Channels)**: The rank was 3 in both configurations. Keeping it constant might be beneficial as it balances compression and error.
- **Mode 2 and Mode 4 (Height and Frames)**: The rank was 9 in the last configuration and 7 in the best configuration. A slight increase might help reduce approximation error.
- **Mode 3 and Mode 4 (RGB Channels and Frames)**: The rank was 2 in both configurations. Increasing this rank could help reduce the approximation error.

**5. Proposed Rank Adjustments:**
- Increase the rank between Mode 1 and Mode 4 from 8 to 10.
- Increase the rank between Mode 2 and Mode 4 from 7 to 8.
- Increase the rank between Mode 3 and Mode 4 from 2 to 3.
**Rank Array**: [12, 3, 10, 3, 8, 3] **Loss Function at iteration 4**: -1.5407013427

---

**Running iteration 5 of iterative prompt**:
**Detailed Reasoning**: **1. Understanding the Current Situation**:
- The current rank array [12, 3, 10, 3, 8, 3] results in a loss function of -1.5407013427.
- The best-known rank array [12, 3, 8, 3, 7, 2] results in a lower loss function of -1.5880439406.
**2. Compression Rate vs. Approximation Error**:
- The current compression rate is 0.1381115846, and the approximation error is 0.0761192149.
- The best-known compression rate is 0.1144228746, and the approximation error is 0.0899020181.
**3. Analyzing the Balance**:
- The current compression rate is higher than the approximation error, suggesting that reducing the ranks could potentially lower the loss function.
- The best-known configuration has a lower compression rate but a slightly higher approximation error, which overall results in a better loss function.
**4. Explorative Adjustments**:
- **Width and Height (Mode 1 and Mode 2)**: These are typically the largest dimensions in video data. A rank of 12 between them seems reasonable, but we could explore reducing it slightly to 10 to see if it impacts the compression rate favorably.
- **Frames (Mode 4)**: Temporal coherence is important, but the ranks involving frames are relatively high. We could try reducing the rank between Mode 1 and Mode 4 from 10 to 8 and Mode 2 and Mode 4 from 8 to 6 to see if it helps reduce the compression rate further without significantly impacting the approximation error.
**5. Proposed Rank Adjustments**:
- Reduce the rank between Mode 1 and Mode 2 from 12 to 10.
- Reduce the rank between Mode 1 and Mode 4 from 10 to 8.
- Reduce the rank between Mode 2 and Mode 4 from 8 to 6.
Based on the above reasoning, the adjusted rank array is:
**Rank Array**: [10, 3, 8, 3, 6, 3]
This configuration aims to balance the compression rate and approximation error more effectively, potentially leading to a lower overall loss function.
**Loss Function at iteration 5**: -1.6246676937

## Time-Series Dataset

**TN Initialization**
**Rank Array**: [3, 2, 2, 1, 3, 2, 1, 2, 1, 1]
**Loss Function at iteration 0**: 1.0711770536

---

**TN Discovery**

**Running iteration 1 of iterative prompt**:
**Detailed Reasoning**:
To refine the rank suggestions and minimize the loss function, we need to carefully balance the compression rate and the approximation error. The current rank array is [3, 2, 2, 1, 3, 2, 1, 2, 1, 1], with a compression rate of 0.2009259259 and an approximation error of 2.7178871524. The loss function is the natural log of the sum of the compression rate and 10 times the approximation error.

**1. Current Situation Analysis**: The compression rate (0.2009) is significantly lower than half of the approximation error (2.7179). This suggests that increasing the ranks could potentially reduce the approximation error more significantly than it would increase the compression rate, leading to a lower loss function.

**2. Strategic Adjustments**:

- Increase Ranks: Since the compression rate is much lower than the approximation error, we should consider increasing some ranks to reduce the approximation error. This could be particularly effective for connections that are currently at their minimum (rank 1), as increasing these could have a substantial impact on reducing the approximation error.

- Targeted Increases: Focus on increasing ranks where the interaction between modes is likely to be more complex and impactful. For instance, connections involving financial instrument types and assets might benefit from higher ranks due to the inherent complexity and variability in financial data.

**3. Proposed Changes**:

- **Increase the rank between Mode 1 and Mode 4** from 1 to 2. This connection might represent a critical interaction between financial instrument types and time points, which could benefit from a higher rank.

- **Increase the rank between Mode 2 and Mode 5** from 1 to 2. This could represent an interaction between assets and indexes of rolling windows, which might capture more variability with a higher rank.

- **Increase the rank between Mode 3 and Mode 5** from 1 to 2. This might capture more detailed interactions between features and indexes of rolling windows.

**Expected Outcome**: These changes are expected to decrease the approximation error significantly while only slightly increasing the compression rate, thus reducing the overall loss function.

**Rank Array**: [3, 2, 2, 2, 3, 2, 2, 2, 2, 1]

**Loss Function at iteration 1**: -0.390709764

---

**Running iteration 2 of iterative prompt**:

**Detailed Reasoning**: Let's take a deep breath and analyze the current situation. The current rank array [3, 2, 2, 2, 3, 2, 2, 2, 2, 1] yields a compression rate of 0.4 and an approximation error of 0.2765764944, resulting in a total loss function of -0.390709764. Our goal is to minimize this loss function by adjusting the ranks, keeping in mind the balance between compression rate and approximation error.

**1. Compression Rate vs. Approximation Error**: The current compression rate (0.4) is relatively low compared to the approximation error (0.2765764944). According to the guideline, if the compression rate is smaller than half of the approximation error, increasing the ranks could potentially decrease the loss function. Here, 0.4 is not smaller than half of 0.2765764944 (which is approximately 0.138), so we don't strictly fall into this category. However, the compression rate is still significantly lower than the approximation error, suggesting that a slight increase in ranks might be beneficial.

**2. Exploration of Rank Changes**: We need to explore both small and large changes in ranks. Since the compression rate is not excessively low compared to the approximation error, we should consider moderate increases in ranks to see if the approximation error can be reduced more significantly than the increase in compression rate.

**3. Mode Interactions**: In financial data, interactions between different modes such as financial instrument types, assets, features, and time points are crucial. Increasing ranks in connections involving these modes might capture more complex interactions, potentially reducing the approximation error.

**4. Proposed Changes**:

- **Increase the rank between (Mode 1, Mode 2)** from 3 to 4. This connection might capture more complex interactions between financial instrument types and assets.

- **Increase the rank between (Mode 1, Mode 5)** from 3 to 4. This could help capture more temporal dynamics across different financial instruments.

- **Increase the rank between (Mode 2, Mode 3)** from 3 to 4. This might improve the interaction modeling between assets and features.

**5. Maintaining Other Ranks**: Keep the other ranks the same to avoid a significant increase in the compression rate, which could offset the benefits of reduced approximation error.
By making these adjustments, we aim to achieve a better balance between the compression rate and approximation error, thereby minimizing the loss function.
**Rank Array**: [4, 2, 2, 4, 4, 2, 2, 2, 2, 1]
**Loss Function at iteration 2**: -0.2087247722

---

**Running iteration 3 of iterative prompt**:
Certainly! Let's take a deep breath and analyze the situation step-by-step.
**Detailed Reasoning**:
**1. Understanding the Current Situation**:
- The current rank array [4, 2, 2, 4, 4, 2, 2, 2, 2, 1] results in a loss function of -0.2087247722.
- The lowest loss function observed was -0.390709764 with the rank array [3, 2, 2, 2, 3, 2, 2, 2, 2, 1].
**2. Loss Function Analysis**:
- The loss function is influenced by both the compression rate and the approximation error.
- The goal is to minimize the sum of the compression rate and 10 times the approximation error.
**3. Compression Rate vs. Approximation Error**:
- If the compression rate is much lower than the approximation error, increasing ranks can help reduce the approximation error significantly, potentially lowering the loss function.
- Conversely, if the compression rate is much higher than the approximation error, reducing ranks can help decrease the compression rate, potentially lowering the loss function.
**4. Current Observations**:
- The current compression rate is 0.7259259259, and the approximation error is 0.085692659.
- The lowest loss function had a compression rate of 0.4 and an approximation error of 0.2765764944.
**5. Strategic Adjustments**:
- The current compression rate is higher than the approximation error, suggesting that reducing ranks could be beneficial.
- The previous lowest loss function had a lower compression rate, indicating that a balance between compression and error is crucial.
**6. Proposed Adjustments**:
- Focus on reducing ranks where the compression rate can be decreased without significantly increasing the approximation error.
- Consider reducing the ranks where the compression rate is disproportionately high compared to the approximation error.
Rank Array Proposal: - **Rank for the connection between (Mode 1, Mode 2)**: Reduce from 4 to 3 to decrease compression rate.
- **Rank for the connection between (Mode 1, Mode 3)**: Keep at 2 as it is balanced.
- **Rank for the connection between (Mode 1, Mode 4)**: Reduce from 4 to 3 to decrease compression rate.
- **Rank for the connection between (Mode 1, Mode 5)**: Reduce from 4 to 3 to decrease compression rate.
- **Rank for the connection between (Mode 2, Mode 3)**: Keep at 2 as it is balanced.
- **Rank for the connection between (Mode 2, Mode 4)**: Keep at 2 as it is balanced.
- **Rank for the connection between (Mode 2, Mode 5)**: Keep at 2 as it is balanced.
- **Rank for the connection between (Mode 3, Mode 4)**: Keep at 2 as it is balanced.
- **Rank for the connection between (Mode 3, Mode 5)**: Keep at 2 as it is balanced.
- **Rank for the connection between (Mode 4, Mode 5)**: Keep at 1 as it is already minimal.
**Final Rank Array**:
**Rank Array**: [3, 2, 3, 3, 2, 2, 2, 2, 2, 1]
This adjusted rank array aims to reduce the compression rate while maintaining a reasonable approximation error, potentially leading to a lower loss function.
**Loss Function at iteration 3**: -0.4233019939

## J EXAMPLE RUNS OF TNLLM IN EACH DATASET

We report the objective value of tnLLM after the initialization and after each step until early stopping for one example run across all 3 datasets.

Table 10: Train and Test Objective Function Values for an example run of tnLLM until early stopping in Images, Videos, and Time-series Datasets.

| Evaluation | Images | | Videos | | Time-series | |
|---|---|---|---|---|---|---|
| | Train | Test | Train | Test | Train | Test |
| TN Initialization | -0.57 | -0.47 | -0.80 | -0.81 | 1.07 | 1.43 |
| TN Discovery 1 | -0.55 | -0.29 | -1.35 | -1.37 | -0.39 | -0.36 |
| TN Discovery 2 | **-0.64** | **-0.50** | -1.59 | -1.66 | -0.21 | -0.20 |
| TN Discovery 3 | - | - | -1.37 | -1.39 | **-0.42** | **-0.42** |
| TN Discovery 4 | - | - | -1.54 | -1.58 | - | - |
| TN Discovery 5 | - | - | **-1.62** | **-1.70** | - | - |

## K FURTHER DATA DOMAINS

We report below the performance of tnLLM against TNLS, TnALE, and tnGPS on three additional datasets from data domains (Physics, Geospatial Hyperspectral, and Biomedical MRI Neuroimaging) beyond the current SOTA TN-SS literature (Li et al., 2022; 2023; Zeng et al., 2024a; Zheng et al., 2024). For the Physics experiment, we use the Darcy Flow dataset from PDEBench (Takamoto et al., 2022). Each tensor has the shape of $128 \times 128 \times 10$, with 10 samples used for training and 10 for testing. For the geospatial hyperspectral image experiment, we use the datasets provided in (Baumgardner et al., 2015) and (Dell'Acqua et al., 2004). Each tensor is cropped to share a common shape of $100 \times 100 \times 80$, where each mode represents the width, height, and spectral bands respectively. For the neuroimaging experiment, we use the BrainWeb MRI dataset (Cocosco, 1997). Each tensor has the shape of $217 \times 181 \times 36 \times 2$, where 217 is the height, 181 is the width, 36 is the number of slices, and 2 is the modality. We use the normal brain data for training and the brain data with multiple sclerosis for testing.

Table 11: Performance comparison across different additional datasets (Physics, Geospatial Hyperspectral, and Biomedical MRI). The values on the left give the lowest training and corresponding testing objective function values. The values in [square brackets] give the number of evaluations required to first achieve the best training objective function value. Best values are bolded and second-best values are underlined.

| Dataset | | TNLS | | TnALE | | tnGPS | | tnLLM | |
|---|---|---|---|---|---|---|---|---|---|
| **Physics** | **Train** | -0.57 | [196] | -0.56 | [162] | -0.55 | [92] | -0.54 | [7] |
| | **Test** | -0.49 | | **-0.56** | | -0.52 | | -0.54 | |
| **Geospatial Hyperspectral** | **Train** | -1.45 | [205] | -1.43 | [151] | -1.43 | [76] | -1.42 | [7] |
| | **Test** | -1.33 | | -1.29 | | **-1.35** | | -1.33 | |
| **Biomedical MRI** | **Train** | -0.46 | [310] | -0.45 | [146] | -0.41 | [130] | -0.48 | [5] |
| | **Test** | -0.54 | | -0.53 | | -0.45 | | **-0.55** | |

