# OpenReview forum: "Domain-Aware Tensor Network Structure Search"
_ICLR.cc/2026/Conference — Submitted to ICLR 2026_

### Official Review · Reviewer_pkqP · 2025-10-30

**Soundness:** 2
**Presentation:** 3
**Contribution:** 3
**Rating:** 4
**Confidence:** 3

**Summary:**

The paper proposes and analyzes LLM prompts to optimize the tensor network structure/rank problem. Informed with domain knowledge on the tensor dimensions, along with evaluations and history the LLM is able to propose network/structures that efficient compressions in a few iterations compared to non-domain informed searches. The model is applied to learn tensor structures for three  datasets of 3rd-order, 4th-order, and 5th order tensors.

**Strengths:**

The paper is clear and interesting. Using an LLM to perform domain-informed optimization seems logical (replaces an expert users intuition and experience).

**Weaknesses:**

The main critique is the the limited amount of data. The two datasets (images and videos) are themselves very related. The other data is interesting but is rather low dimensional. Overall, for a method that should use domain knowledge to inform, a more diverse set of datasets from various domains should be tested to provide evidence to support the claims.

**Questions:**

Have tensor network compression for physics data been used?

Geospatial hyperspectral imaging across multiple time points (change detection) could provide rich data.

How about biomedical data?

How about dynamic graph compression? Especially multiple graph layers or hyper-edges across time can be 3rd, 4th order tensors.

Neuroimaging (MRI) with longitudinal subjects can be organized in space (3D), as well as multiple visits per subject and multiple subjects.

 Diffusion tensor imaging also has even higher-order?

---

> ### Author Response · Authors · 2025-11-21
>
> We thank the reviewer for their effort and valuable feedback. We believe there may have been some misunderstanding. Please find our answers to your comments and questions below.
>
> > **Weakness 1**: “The two datasets (images and videos) are themselves very related. The other data is interesting but is rather low dimensional”
>
> We thank the reviewer for their feedback. We believe there may be a misunderstanding here. We have followed the practice of existing TN-SS literature in choosing data domains to validate our proposed method [1-4], which primarily included machine learning datasets (Computer-Vision (CV) related). We respectfully clarify a potential misunderstanding regarding our time-series dataset. As stated in Section 4, the time series dataset is an order-5 tensor, making it higher order than the images and videos datasets. We curated this custom financial dataset to demonstrate our method’s generalizability beyond the regular machine learning datasets. The time-series dataset has complex mode relationships which are highly technical and domain-specific, and is the largest dataset in terms of number of samples in the TN-SS literature.
>
> ---
>
> > **Questions** Regarding Tensors from Other Fields
>
> Thank you very much for these excellent suggestions for future applications. We have followed the practice of existing TN-SS literature in choosing data domains to validate our proposed method. The domains mentioned in the questions are currently unexplored in the SOTA TN-SS literature (e.g., TnALE[1], TNLS[2], SVDinsTN[3], tnGPS[4]). However, we agree that data from fields like physics, neuroimaging, and geospatial are exciting for future TN-SS research.
>
> [1] Chao Li, Junhua Zeng, Chunmei Li, Cesar Caiafa, and Qibin Zhao. Alternating local enumeration (TnALE): solving tensor network structure search with fewer evaluations. In Proceedings of the 40th International Conference on Machine Learning, ICML’23. JMLR.org, 2023.
>
> [2] Chao Li, Junhua Zeng, Zerui Tao, and Qibin Zhao. Permutation search of tensor network structures via local sampling. In Proceedings of the 39th International Conference on Machine Learning, pages 13106–13124, Jul 2022.
>
> [3] Yu-Bang Zheng, Xi-Le Zhao, Junhua Zeng, Chao Li, Qibin Zhao, Heng-Chao Li, and Ting-Zhu Huang. SVDinsTN: A tensor network paradigm for efficient structure search from regularized modeling perspective. In Proceedings of the IEEE/CVF Conference on Computer Vision and Pattern Recognition, 2024.
>
> [4] Junhua Zeng, Chao Li, Zhun Sun, Qibin Zhao, and Guoxu Zhou. tnGPS: Discovering unknown tensor network structure search algorithms via large language models (LLMs). In Proceedings of the Forty-first International Conference on Machine Learning, 2024.
>
> ---
>
>
> Given that the reviewer’s main critique concerns the “limited amount of data,” which we believe stems from a misunderstanding as we use the same datasets as in existing SOTA TN-SS papers [1–4] accepted in top-tier AI conferences, we would like to kindly ask the reviewer to reconsider and potentially increase their rating.

---

> > ### Comment · Reviewer_pkqP · 2025-11-26
> >
> > The 5th order example is great but the number of dimensions per mode is quite small. The fact that the sample size is highest is interesting.
> >
> > In my opinion there is a fundamental difference in the numerical optimization methods and the LLM based optimization as it is using prior knowledge for an informed combinatorial search, rather than purely sample evaluation driven. Thus, a larger number of tensor problems of various domains are needed to evidence the claims for LLM than for other hypothesis free approaches. Given the novelty of LLM approach, I think it is reasonable to go beyond existing work.
> >
> > Additionally, what guarantees do we have that the LLM didn't read these exact papers, retrieving solutions that work well from them.  In that case additional novel datasets are absolutely necessary. To me this seems like a much more powerful tool compared the datasets it is tested (besides the 5th order one). To justify claims I believe more examples from different domains are needed.
> >
> > One approach to create more datasets that be to remove modes from the constructed data (i.e., only use one type of financial asset, use only one type of feature per asset, use only one window..)

---

> ### Author Response · Authors · 2025-11-27
>
> Thank you very much for your continued engagement and valuable suggestions regarding the additional domains, which allow us to further highlight the advantages of the proposed tnLLM framework. We will include the results of additional domains in the updated manuscript.
>
> Based on your suggestions, we have conducted additional experiments on 1) **physics**, 2) **biomedical neuroimaging (MRI)**, and 3) **geospatial hyperspectral** data. For the physics experiment, we use the Darcy Flow dataset from PDEBench [1]. Each tensor has the shape of 128 × 128 × 10, with 10 samples used for training and 10 for testing. For the neuroimaging experiment, we use the BrainWeb MRI dataset [2]. Each tensor has the shape of 217 × 181 × 36 × 2, where 217 is the height, 181 is the width, 36 is the number of slices, and 2 is the modality. We use the normal brain data for training and the brain data with multiple sclerosis for testing. For the geospatial hyperspectral image experiment, we use the datasets provided in [3] and [4]. Each tensor is cropped to share a common shape of 100 × 100 × 80, where each mode represents the width, height, and spectral bands respectively.
>
> The results are summarized in the Table below. The values on the left give the lowest training and corresponding testing objective function values. The values in [square brackets] give the number of evaluations required to first achieve the best training objective function value. As observed, the proposed tnLLM is able to achieve superior or comparable objective values while having significant speed ups in  terms of number of evaluations (up to 62x compared to TNLS, 29x compared to TnALE, and 26x compared to tnGPS) in all three additional domains.
>
> | Dataset |   | TNLS | TNALE | TNGPS | TNLLM |
> | :--- | :--- | :--- | :--- | :--- | :--- |
> | **Physics** | Train | -0.57 &nbsp; [196] | -0.56 &nbsp; [162] | -0.55 &nbsp; [$\underline{92}$] | -0.54 &nbsp; [**7**] |
> | | Test | -0.49 | **-0.56** | -0.52 | $\underline{-0.54}$ |
> | **Geospatial Hyperspectral** | Train | -1.45 &nbsp; [205] | -1.43 &nbsp; [151] | -1.43 &nbsp; [$\underline{76}$] | -1.42 &nbsp; [**7**] |
> | | Test | $\underline{-1.33}$ | -1.29 | **-1.35** | $\underline{-1.33}$  |
> | **Biomedical MRI** | Train | -0.46 &nbsp; [310] | -0.45 &nbsp; [146] | -0.41 &nbsp; [$\underline{130}$] | -0.48 &nbsp; [**5**] |
> | | Test | $\underline{-0.54}$ | -0.53 | -0.45 | **-0.55** |
>
> Indeed, the time-series dataset is specifically curated to ensure that it is not part of the LLM's training data and not present in any existing TN-SS works (see lines 320–321 in the manuscript). Moreover, as shown in Table 1 and the table above, our proposed framework is capable of finding improved structures than the ones found using the existing methods.
>
> [1] Makoto Takamoto, Timothy Praditia, Raphael Leiteritz, Dan MacKinlay, Francesco Alesiani, Dirk Pflüger, and Mathias Niepert. PDEBench: An Extensive Benchmark for Scientific Machine Learning. In Proceedings of the 36th Conference on Neural Information Processing Systems Datasets and Benchmarks Track, 2022.
>
> [2] Chris A. Cocosco, Vaskev Kollokian, Remi K.-S. Kwan, and Bruce G. Pike. BrainWeb: Online Interface to a 3D MRI Simulated Brain Database. NeuroImage, vol. 5, no. 4, part 2/4, S425, 1997.
>
> [3] Marion F. Baumgardner, Larry L. Biehl, and David A. Landgrebe. 220 Band AVIRIS Hyperspectral Image Data Set: June 12, 1992 Indian Pine Test Site 3. Purdue University Research Repository, 2015.
>
> [4] Fabio Dell'Acqua, Paolo Gamba, Alessandro Ferrari, J. A. Palmason, Jon Atli Benediktsson, and K. Arnason. Exploiting spectral and spatial information in hyperspectral urban data with high resolution. In IEEE Geoscience and Remote Sensing Letters, vol. 1, no. 4. IEEE, 2004.

---

### Official Review · Reviewer_o65j · 2025-10-31

**Soundness:** 1
**Presentation:** 3
**Contribution:** 2
**Rating:** 2
**Confidence:** 4

**Summary:**

The paper proposes the use of large language models (llm) to speed up the tensor network structure search (TN-SS). In particular it improves upon sampling based algorithms, by letting an llm predict a good initialization structure as well as picking a good candidate during the local neighbourhood search / sampling. In a sense the llm replaces a otherwise complex hardcoded heuristic in the optimization loop, automating human reasoning. In the experiments the paper demonstrates that this indeed successful over three different domains. However, I see some serious issues regarding related work and the fairness of the experiments as well as the general reproducability, as I will detail the weaknesses section.

**Strengths:**

- The idea to let an llm guide the discrete optimization is interesting. While this is not the first work in this general direction, the proposed method for TN-SS is novel. While some may consider the mindest "throwing an llm at the problem" to be uninspiring, I believe it is important to thoroughly test und understand the limits of llms nonetheless. Replacing complex heuristics with concise prompts and llm reasoning has potential
- The experiments show a quick convergence to good solutions with very few calls to the optimization objective on three different domains. This is important since calculating the objective function can be costly.
- Overall the paper is well written and structured
- Tested the results with different models and show good end results across models.

**Weaknesses:**

In its current form the paper has several weaknesses with regard to the related work and the experiments, which overall do not meet the standards for a publication at a top conference like ICLR. However, I believe most of them can be addressed and I will raise my score if this is done properly. I will go over them according to my priorities.

Major Weaknesses:
1. The state of the art seems less clear than claimed in the paper. In particular, SVDinsTN claims to be much faster than TNLS and TnALE, while achieving similar compression quality and Guo et al. claim a speed and quality advantage over TnALE. However, these methods, even though they are cited, are neither properly discussed in the related work section nor are they included as baselines in the experiments. Since, SVDinsTN only seems to require a single pass over the data and still gives strong results in their evaluation it seems to be a perfect fit for initializing TnALE and TNLS. On the other hand, the program synthesis approach of Gua at el. improves upon the number of function evaluations among other things. Thus, for a fair state of the art comparison these two methods should be included in the experiments. It is also crucial to understand if data driven methods like SVDinsTN, might still provide the better initializations than the learned bias of llms. Which leads me to my next weakness.

> Possible changes:
> - Include all SOTA methods as baselines in the experiments. You could probably remove TNLS as the oldest method, if needed.
> - If this is really not possible, remove the many claims about SOTA performance. And explain that there are different approaches to the problems you address already. You could refocus the paper to just evaluating if llms are useful for this task at all, and clearly show that this is indeed the case. As stated by ICLR you do not need to beat the SOTA to get accepted.


2. The paper proposes a complete pipeline, but for me a major part of the contribution would be analyzing the two central parts, initialization and local optimization (TN discovery), independently. Right now it is entirely unclear how useful each part is on its own and thus if llms are actually useful in replacing local search heuristics *directly* in this setting. However, there are no samples for the results of the output of the llm during TN discovery and only final results after both steps. Providing results after the initialization and also showing the convergence behaviour over different runs would be essential for the paper. Additionally, this would allow a direct comparison with SVDinsTN for initialization.

> Possible changes:
> - Report the objective value of tnLLM after the initialization and ideally after each step in the appendix (since these are mostly less than 10, this should be easily doable). The data for this should be already there. Visaulizing this convergence behaviour of tnLLM in a plot would be really interesting, and should at least be included in the appendix.
> - Provide examples of the llm output for the TN discovery parts
> - To better judge the impact of the llm initialization and optimization loop independently, the experiment from Figure 7 could be repeated with less steps of tnLLM, ideally once with only the suggested initialization.
> - Report results for SVDinsTN as comparison for llm initialization. I am aware that this might requiring significant extra work, but I believe its important to fairly assess the performance of llms vs. classic approaches. You can also do the comparison after the first optimization pass, since SVDinsTN also needs one.

3. The comparisons in the experiment do not seem to be entirely fair. The baselines are initialized as "fully-disconnected" graph, while tnLLM starts with a complete graph, which seems to be much closer to the optimum. The llm, never seems to predict 1 which would remove an edge from the graph (it might have for time series dataset, since Table 6 misses several mode pairs). For a fairer comparison the baselines could also start with a complete graph. Finally, the graphs in Figure 7 show, that TNLS and especially TnALE are really close to the final objective function value early on and then seem to iterate for a long time without any visible improvement. This suggests to me that they could report convergence much earlier than they do right now leading to much fewer function evaluations.

> Possible changes:
> - Initialize baselines with a fully connected graph using low or medium sized ranks, could also be randomized or uniform.
> - Clearify the convergence behaviour of the baselines, consider introducing a threshold that stops the iteration if the changes in the objective value are very small for several iterations.

4. All algorithms in the paper are randomized, therefore each experiment should be repeated several (>=5) times and the results should include error bars in plots or report the median and mean average deviation in tables (or alternatively mean and std). However, this is currently only done for tnLLM and as it seems not even there in all experiments. Considering the longest run time is just over a day and most are only a few hours, this seems feasible until the author reviewer discussion period. Even if not, I would already be happy with some partial data for the faster cases and the full data until the final publication. Moreover, improving on 3. might help with reducing the time spent here.

5. The paper claims to include code in the abstract, but the code in the provided supplemental material seems to be incomplete and comes without any documentation. It seems to me that the matlab files that would actually evaluate the objective functions are missing. All submitted artifacts should be up to the standard of ICLR, for me that means code with clearly defined dependencies and a documented straightforward setup. This would greatly increase the reproducibility and also usefulness of the submission to the machine learning community. Ideally, the code does not only include the proposed method, but also all code used to run the experiments.

> For 4. and 5. the required changes should be clear and are necessary for a publication at a top conference from my perspective.

Less important weaknesses:
 - The introduction mentions several use cases for tensor networks in general, but use cases for tensor network structure search specifically would be more relevant.
- Using an llm for optimization *directly* has been studied in a ICLR 2024 paper: https://openreview.net/forum?id=Bb4VGOWELI, this seem like relevant related work to me. I initially thought this would be a novelty of this paper.
 - The paper should report the relative error and compression ratio along the objective function as it was done in previous works from the field, because they are much easier to understand by humans.
- Some cited works in the TN-SS section are of little relevance. The paper from Meirom et al. is about contraction path optimization which has nothing to do with TN-SS. The work from Chen et al. is only about tensor rings and trains, thus the structure is mostly given already. Consider removing these citations, especially the first one.
- The claim that the timeseries dataset "is the largest dataset in terms of number of samples ever considered in the
TN-SS problem" is unecessary and hard to verify. Moreover, its still only about 1,2 MB large, so in terms of modern data and actual real world applications it is tiny and overall smaller than the video dataset. I would just remove the claim, it does not add anything to your work.

Some minor things (irrelevant for the score):
- The link to the data is broken, they probably changed something on their page: http://trace.eas.asu.edu/yuv/
- The related work on TN-SS is basically repeated twice, once in the intro and then again in related work, I think it could be shortened in the intro
- I am not convinced that the explanations by the llm are very useful for *domain experts*, however they can be useful for debugging just like any other runtime artifact. It seems similar to stepping through a hardcoded heuristic in a debugger.

**Questions:**

- Why do you take the natural logarithm of the usual objective function in TN-SS in equation (1)
- Why are so many mode pairs missing in table 6? According to your own prompt there should be 10, not 4
- Algorithm 1:  Why you always check the whole history P for a better Graph and not only the last one, i.e. H? When does the Algorithm converge?
- In Line 179-180 there is a single Tensor X, in equation (1) there are L tensors, why the switch?
- Since you run tnLLM ten times in the hybrid algorithm, why is the objective still so bad at the beginning in Figure 7, how does that relate to the results in Table 1? There it seems 10 runs are almost always sufficient for achieving -0.41.

---

> ### Author Response · Authors · 2025-11-21
>
> Thank you for the thorough feedback and acknowledging the novelty and strengths of our work. Please find our answers to your comments and questions below.
>
> ---
>
> > **Major Weakness 1**: “SVDinsTN claims to be much faster than TNLS and TnALE… Since, SVDinsTN only seems to require a single pass over the data and still gives strong results in their evaluation it seems to be a perfect fit for initializing TnALE and TNLS…”
>
> We did not compare with SVDinsTN and the method by Guo et al. because their methods are designed for different experimental settings. In our experiments, we try to find generalizable TN structures to unseen test tensors by only using the training tensors. SVDinsTN and the method by Guo et al. are designed to find the TN structure of a single tensor, rather than the generalizable TN structure of a training set of tensors.
>
> Moreover, the same optimization solver is used in our experiments across TNLS, TnALE, tnGPS and the proposed tnLLM, making the number of evaluations a fair comparison metric between methods. This optimization solver takes in the identified TN structures and outputs their corresponding objective function values in the entire training/test datasets. Since SVDinsTN itself is essentially a different optimization solver which searches for the TN structure of a single tensor and is quite sensitive to hyperparameter choices, its ‘single pass of the data’ is more time consuming and defined to be a single pass of a single tensor, rather than a single pass of the entire training dataset of tensors.
>
> ---
>
> > **Major Weakness 2**: “A major part of the contribution would be analyzing the two central parts, initialization and local optimization (TN discovery), independently. … Providing results after the initialization and also showing the convergence behaviour over different runs would be essential for the paper. Additionally, this would allow a direct comparison with SVDinsTN for initialization.”
>
> Thank you for pointing out these points which can strongly strengthen our work. As requested, we now
>
> 1. Report the objective values of tnLLM after initialization and after each step until early stopping for one example run across all 3 datasets in Table 10 (below) of Appendix J in the revised manuscript.
>
> |   Evaluation | Images (Train)   | Images (Test)   |   Videos (Train) |   Videos (Test) | Time-series (Train)   | Time-series (Test)   |
> |-----:|:-------|:------|---:|---:|:-----|:---|
> | TN-Initialization | -0.57 | -0.47 |  -0.80 | -0.81 | 1.07    | 1.43  |
> | TN-Discovery 1 | -0.55 | -0.29 |   -1.35 | -1.37 | -0.39   | -0.36 |
> | TN-Discovery 2 | -0.64 | -0.50 |   -1.59  | -1.66 | -0.21    | -0.20 |
> | TN-Discovery 3 | - | -  | -1.37 |  -1.39 | -0.42   | -0.42    |
> | TN-Discovery 4 | - | - |-1.54 | -1.58 | -  | - |
> | TN-Discovery 5 | -  | -  | -1.62 | -1.70 | -  | - |
>
> 2. Provide examples of tnLLMs explanations during the TN discovery stage for all 3 datasets in Appendix I.
>
> As tnLLM takes very few evaluations to complete its search, we directly utilized its outputs in the experiments of Figure 7. More specifically, the objective function value progression of tnLLM (the first 10 evaluations) are also already included in Figure 7. Furthermore, SVDinsTN finds the TN structure of a single tensor, so it’s not applicable to our experiment setting.

---

> ### Author Response · Authors · 2025-11-21
>
> ---
>
> > **Major Weakness 3**: “The comparisons in the experiment do not seem to be entirely fair. The baselines are initialized as "fully-disconnected" graph, while tnLLM starts with a complete graph, which seems to be much closer to the optimum. The llm, never seems to predict 1 which would remove an edge from the graph … Finally, the graphs in Figure 7 show, that TNLS and especially TnALE are really close to the final objective function value early on and then seem to iterate for a long time without any visible improvement.”
>
> We would like to clarify that tnLLM is not differently initialized from other baselines to be a complete graph. Indeed, the starting point of tnLLM is automatically found by its TN initialization process.
>
> Furthermore, the TN structures produced by tnLLM do contain ‘1’s in the edge connectivities (See for example Table 7 (originally Table 6): Mode 4 & Mode 5 in run 2).
>
> Regarding convergence, we have followed the suggestions by the authors in their original manuscripts of the baselines. However, as requested, we have additionally added the table below as Table 9 in Appendix H of the revised manuscript to report the convergence behaviour of the baselines with an early stopping threshold of 0.01 in the objective function with patience 15. As observed, the proposed tnLLM is still able to achieve significant speed ups in number of evaluations (up to 68x compared to TNLS, 24x compared to TnALE, and 53x compared to tnGPS), while the objective values achieved by the baselines are now slightly worse relative to those reported in Table 1 of the original manuscript.
>
> | **Data Type** |        | **TNLS** (Value / [Evals])              | **TnALE** (Value / [Evals])                  | **tnGPS** (Value / [Evals])                 | **tnLLM (Ours)** (Value / [Evals])         |
> |-----|--------|-------------|------|---------|-----|
> | **Images**    | **Train** | -0.64 ± 0.01  / [58.6 ± 5.2]            | -0.63 ± 0.02 / [51.2 ± 10.4]               | -0.64 ± 0.01 / [210.0 ± 22.3]               | -0.63 ± 0.01 / [**4.0** ± 1.9]              |
> |               | **Test**  | -0.46 ± 0.02                         | -0.46 ± 0.01                               | -0.42 ± 0.03                                | **-0.48** ± 0.05                            |
> | **Videos**    | **Train** | -1.63 ± 0.02 / [422.2 ± 20.1]           | -1.66 ± 0.01 / [148.8 ± 11.5]              | -1.65 ± 0.01 / [185.0 ± 26.8]               | -1.63 ± 0.01 / [**6.2** ± 3.5]              |
> |               | **Test**  | -1.70 ± 0.03                            | **-1.72** ± 0.01                             | -1.71 ± 0.02                               | -1.70 ± 0.02                                |
> | **Time-series** | **Train** | -0.46 ± 0.01 / [170.6 ± 7.6]           | -0.46 ± 0.03 / [115.2 ± 14.4]                | -0.40 ± 0.02 / [62.6 ± 17.0]              | -0.42 ± 0.02 / [**5.6** ± 4.4]              |
> |               | **Test**  | -0.42 ± 0.01                          | **-0.44** ± 0.02                            | -0.40 ± 0.02                                | -0.41 ± 0.02                                |
>
>
>
> Further, as requested, we tried initializing the baseline methods (TNLS and TnALE) with randomly sampled ranks and observed that they require significantly longer run time due to higher ranks in some edges and the high computational complexity of the fully connected tensor network. For example, one evaluation (a single pass of the training data) in the time-series dataset took 8 hours to run. Putting this into context, TnALE originally required around 22.4 hours in the time-series dataset for the full search of 177 evaluations. Thus, initializing the baselines as ‘fully disconnected’ graphs is more favorable to them.
>
> ---
>
> > **Major Weakness 4**: “... each experiment should be repeated several (>=5) times and the results should include error bars in plots or report the median and mean average deviation in tables (or alternatively mean and std).”
>
> Thank you for pointing out this change which will further strengthen our analysis. As requested, the mean and standard deviations over 5 runs of the performances of all methods have now been included in the table above and in Table 9 of Appendix H in the revised manuscript (with the early stopping criteria implemented for baselines).
>
> ---
>
> > **Major Weakness 5**:  “The paper claims to include code in the abstract, but the code in the provided supplemental material seems to be incomplete and comes without any documentation. … Ideally, the code does not only include the proposed method, but also all code used to run the experiments.”
>
> Thank you for pointing this out. Yes, the current code includes the implementation of the proposed tnLLM framework. We are happy to share the full code for the optimization solver and the datasets. Please see the anonymised link to our code in another post below.

---

> > ### Comment · Reviewer_o65j · 2025-11-21
> > **Re Weakness 3**
> >
> > Thanks, I still think that some more sensible initialization for the baselines might be possible (instead of entirely random), but I appreciate the effort you put into this point. As the sensible initialization might depend on the domain, your approach does seem reasonbly verified now.

---

> ### Author Response · Authors · 2025-11-21
>
> > **Minor Weakness 1**: “The introduction mentions several use cases for tensor networks in general, but use cases for tensor network structure search specifically would be more relevant.”
>
> Thanks for pointing this out. We believe that the use cases of tensor network structure search lies in the real-world adoption of any general tensor network application, which currently lacks a systematical way of deciding on the tensor network structures. Therefore, we believe that the advancement of TN-SS methods promises to make tensor network methods more applicable and prominent in general.
>
> ---
>
> > **Minor Weakness 2**: “Using an llm for optimization directly has been studied in a ICLR 2024 paper: https://openreview.net/forum?id=Bb4VGOWELI, this seem like relevant related work to me. I initially thought this would be a novelty of this paper.”
>
> Thank you for bringing this paper to our attention. We now cite it in the related works (Section 1.1). However, our paper’s core contribution lies in the integration of domain information into the TN-SS problem, rather than simply using an LLM for optimization, whose performance we have examined in the ablation study of Section 4.4.
>
> ---
>
> > **Minor Weakness 3**: “The paper should report the relative error and compression ratio along the objective function as it was done in previous works from the field, because they are much easier to understand by humans.”
>
> We thank the reviewer for this suggestion regarding making the optimization results more readable to the readers. Since TN-SS in compression applications is usually used to find the most compressed TN format given an error threshold, we report here the best compression rate (number of entries in the original tensor/ number of entries in its TN compressed format) achieved by all methods for a given test approximation error threshold (0.02 for Images, 0.01 for Videos and 0.01 for Time-Series dataset). An error threshold of 0.01 can be interpreted as almost a ‘perfect’ reconstruction. We take 0.02 as the test approximation error threshold for the Images dataset, as none of the methods achieved an approximation error lower than 0.01. A larger compression rate indicates more compression. tnLLM is observed to achieve the best compression rates in two of the three datasets.
>
> | | TNLS | TnALE | tnGPS | tnLLM |
> | :--- | :--- | :--- | :--- | :--- |
> | Images | 2.54 | 2.25 | 2.47 | **2.63** |
> | Videos | 12.38 | **12.67** | 11.52 | 11.22 |
> | Time-Series | 1.47 | Failed | 1.49 | **1.60** |
>
> We have also added this table as Table 2 in Section 4 of the revised manuscript.
>
> ---
>
> > **Minor Weakness 4**: “Some cited works in the TN-SS section are of little relevance. The paper from Meirom et al. is about contraction path optimization which has nothing to do with TN-SS. The work from Chen et al. is only about tensor rings and trains, thus the structure is mostly given already. Consider removing these citations, especially the first one.”
>
> The intermediate steps in tensor contraction path optimization can be thought of as different TN structures (after intermediate contractions), so although not directly comparable as a baseline, we have included it due to its relevance. The work from Chen et al. is about tensor rings and trains, but they also consider the different permutations of mode indices, which can be seen as a sub-task of TN-SS.
>
> ---
>
> > **Minor Weakness 5**: “The claim that the timeseries dataset "is the largest dataset in terms of number of samples ever considered in the TN-SS problem" is unecessary and hard to verify.”
>
> The time-series dataset was curated to test the generalizability of the identified TN structures, in which the generalization becomes much harder when the number of samples in a dataset increases. The number required in the optimization solver also increases linearly with the number of samples. Therefore, we believe including this point is important.

---

> ### Author Response · Authors · 2025-11-21
>
> > **Question 1**: “Why do you take the natural logarithm of the usual objective function in TN-SS in equation (1)”
>
> Taking the natural logarithm does not affect the results of the baselines and proposed method. It simply scales the objective function values to a range easier for visualization in plots.
>
> ---
>
> > **Question 2**: “Why are so many mode pairs missing in table 6? According to your own prompt there should be 10, not 4”
>
> We have revised Table 6 (Table 7 in revised manuscript) to include all 10 modes across all 3 runs.
>
> ---
>
> > **Question 3**: “Algorithm 1: Why you always check the whole history P for a better Graph and not only the last one, i.e. H? When does the Algorithm converge?”
>
> This is how sampling-based TN-SS algorithms are defined in prior work. We agree that just checking the latest set of graphs would be more efficient, and we have already implemented our code this way. These algorithms stop when encountering a time-limit or with an annealing learning rate which shrinks the local search space gradually.
>
> ---
>
> > **Question 4**: “In Line 179-180 there is a single Tensor X, in equation (1) there are L tensors, why the switch?”
>
> Line 179-180 serves the purpose to define the notation of a tensor. In the experiments, we have multiple (e.g., L) tensors, for which we aim to find a generalizable TN structure for the test set using the TN-SS algorithms and the training set of tensors.
>
> ---
>
> > **Question 5**: “Since you run tnLLM ten times in the hybrid algorithm, why is the objective still so bad at the beginning in Figure 7, how does that relate to the results in Table 1? There it seems 10 runs are almost always sufficient for achieving -0.41.”
>
> In Figure 7, the first 10 evaluations plotted correspond to the TN initialization + 9 iterations of TN discovery in tnLLM. Therefore, the results in Table 1 corresponds to the objective function values at Evaluation index 10 in Figure 7.
>
> After this “global search’’ using tnLLM, we continue with TNLS or TnALE. The hybrid algorithm can further improve the objective function value found by tnLLM, reaching -0.44 (see Table 8 (originally 7) of the revised manuscript ) within ~20 evaluations in total (including both tnLLM + TNLS/TnALE evaluations) for the time series dataset.
>
> ---
>
> Thank you very much for your valuable suggestions regarding the minor things. We will improve the paper accordingly in the camera ready version.
>
> ---
>
> We believe we have answered all your questions and concerns. We hope that you will consider increasing your score. Please let us know if any issues remain and we are happy to address them.

---

> > ### Author Response · Authors · 2025-11-21
> >
> > We thank the reviewer for the detailed feedback and the suggestions which we believe have improved the quality of our paper. We list below the additions in the revised manuscript in relation to the reviewer's suggestion for clarity. All additions in the revised manuscript are in blue.
> >
> > > **Major Weakness 2**
> >
> > We have reported the objective value of tnLLM after initialization and after each step until early stopping for one run across all 3 datasets in Table 10 of Appendix J in the revised manuscript.
> >
> > We have provided examples of tnLLMs explanations during the TN discovery stage for all 3 datasets in Appendix I.
> >
> > ---
> >
> > > **Major Weakness 3 and 4**
> >
> > We have added Table 9 in Appendix H of the revised manuscript to report the convergence behaviour of the baselines using an early stopping threshold of 0.01 on the objective function, with a patience of 15. The mean and standard deviation for all methods and datasets over 5 runs are reported to address Major Weakness 4.
> >
> > ---
> >
> > > **Minor Weakness 2**
> >
> > We  have added Table 2 in Section 4 of the revised manuscript in order to report the the best compression rate (number of entries in the original tensor/ number of entries in its TN compressed format) achieved by all methods for a given test approximation error threshold.
> >
> > ---
> >
> > > **Minor Weakness 6**
> >
> > We have updated Table 6 (Table 7 in revised manuscript) to include the explanations for the initialized TN structure across all 10 modes and 3 runs of the times-series dataset.

---

> > > ### Author Response · Authors · 2025-11-21
> > >
> > > Please find the anonymous link to the full code including the proposed method, optimization solver and data as requested by the reviewer:
> > >
> > > https://anonymous.4open.science/r/tnLLM-B204/README.md

---

> > ### Comment · Reviewer_o65j · 2025-11-21
> > **Re Question 1**
> >
> > So of course the logarithm does not change the point of optimality, but if you just did it for the visualizations, you should have  used a log scaled axis, instead of changing the objective function.

---

> ### Comment · Reviewer_o65j · 2025-11-21
> **Re Weakness 1**
>
> Thanks for your detailed answers, I will focus on the points in your answers I still disagree on for now and go over all your changes in the next week.
>
> TnALE and TNLS use the same framing with a single tensor in their respective papers and yet you still include them as a baselines. So it remains unclear to me why they should not be included in a comparison claiming to cover the state of the art.
> Afterall, a series of tensors can be thought of a single tensor with an additional axis.

---

> ### Comment · Reviewer_o65j · 2025-11-21
> **Re Weakness 4 & 5**
>
> Re W4:
>
> TN-SS and contraction path optimization have nothing in common apart from both working with tensor networks. The optimization objective is entirely different both semantically as well as syntactically. It absolutely does not fit into the list of works you cite in the context as extracted from your paper below:
>
> > Various approaches have been proposed to address the TN-SS problem, including Bayesian inference (Zeng et al., 2024b), spectrum methods (Chen et al., 2024), reinforcement learning (Meirom et al., 2022), program synthesis (Guo
> et al., 2025), and continuous optimization (Zheng et al., 2024).
>
> Re W5:
> I myself ran tensor decompositions on the MNIST dataset, which is tiny by your modern standard, but still much larger than your time series dataset. By your relatively loose definition of related works above, finding the rank of these decompositions counts as basic structure search.

---

> ### Author Response · Authors · 2025-11-23
> **Re Re Major Weakness 1 & 3, Question 1, Minor Weakness 4**
>
> Thank you very much for your prompt reply. We appreciate the continued engagement and the opportunity to clarify these technical details. Please see our responses below.
>
> > **Re: Weakness 1 (SVDinsTN applicability to sets of tensors)**: So it remains unclear to me why SVDinsTN should not be included in a comparison...
>
> While a set of tensors can indeed be stacked into a single tensor with an additional axis, applying SVDinsTN to this stacked tensor will not achieve a meaningful compression. We provide the mathematical illustration below for a set of order-3 RGB image tensors ($\mathcal{Y} \in \mathbb{R}^{H \times W \times C}$). We demonstrate that attempting to find a generalizable order-3 structure on a stacked order-4 tensor using SVDinsTN results in a trivial compression where all images become scalar multiples of a single image.
>
> Let $\mathcal{X} \in \mathbb{R}^{H \times W \times C \times N}$ be an order-4 tensor representing a stack of $N$ order-3 RGB images. Modes $1,2,3$ correspond to Height, Width, RGB channels, and mode $4$ corresponds to the sample index $N$. To find a generalizable order-3 TN structure applicable to individual order-3 tensors, in the SVDinsTN formulation, it requires setting the ranks connected to the last mode to 1, i.e., $R_{1,4} = 1, R_{2,4} = 1, R_{3,4} = 1$ (see [1], definition (1)).
>
> The order-4 SVDinsTN element-wise definition for $\mathcal{X}$ under these constraints becomes:
> $$
> \mathcal{X}(h, w, c, n) = \sum\_{r\_{1,2}=1}^{R\_{1,2}} \sum\_{r\_{1,3}=1}^{R\_{1,3}} \sum\_{r\_{2,3}=1}^{R\_{2,3}} \mathbf{S}\_{1,2}(r\_{1,2},r\_{1,2}) \mathbf{S}\_{1,3}(r\_{1,3}, r\_{1,3}) \mathbf{S}\_{2,3}(r\_{2,3}, r\_{2,3}) \mathbf{S}\_{1,4}(1, 1) \mathbf{S}\_{2,4}(1, 1) \mathbf{S}\_{3,4}(1, 1) \mathcal{G}\_1(h, r\_{1,2}, r\_{1,3}, 1) \mathcal{G}\_2(r\_{1,2}, w, r\_{2,3}, 1) \mathcal{G}\_3(r\_{1,3}, r\_{2,3}, c, 1) \mathcal{G}\_4(1, 1, 1, n)
> $$
>
> $\mathbf{S}_{t,l}$ represent the diagonal factors representing the edges between modes, and $\mathcal{G}_k$ are the TN cores. By absorbing all factors independent of $n$ into a base tensor $\mathcal{Z}$, we obtain:
>
> $$
> \mathcal{X}(h, w, c, n) = \mathcal{Z}(h, w, c, 1, 1, 1) \mathcal{G}_4(1, 1, 1, n)
> $$
>
> Dropping the modes whose size are 1, this simplifies to:
>
> $$
> \mathcal{X}(:, :, :, n) = \mathcal{Z}(:, :, :) \cdot \mathbf{g}_4(n)
> $$
>
> where $\mathbf{g}_4(n)$ is a scalar given a specific $n$. Consequently, SVDinsTN can only compress the dataset meaningfully if every image is just a scalar multiple of the same base tensor $\mathcal{Z}$, which is insufficient for meaningful structure search on complex datasets.
>
> [1] Yu-Bang Zheng, Xi-Le Zhao, Junhua Zeng, Chao Li, Qibin Zhao, Heng-Chao Li, and Ting-Zhu Huang. SVDinsTN: A tensor network paradigm for efficient structure search from regularized modeling perspective. In Proceedings of the IEEE/CVF Conference on Computer Vision and Pattern Recognition, 2024.
>
>
> > **Re: Weakness 3 (Initialization)**: ...As the sensible initialization might depend on the domain, your approach does seem reasonbly verified now.
>
> We thank the reviewer for the insightful analysis and for acknowledging the validity of our approach.
>
>
> > **Re: Question 1 (Use of Logarithm)**: the logarithm does not change the point of optimality...
>
> We appreciate this valuable suggestion. We agree that using the logarithm does not affect the outcome of the compared algorithms. We will also further explicitly clarify that this step is optional in the camera-ready version.
>
> > **Re: Minor Weakness 4 (References)**: TN-SS and contraction path optimization have different...
>
> Thank you very much for the clarification. We have now removed these references and replaced them with another earlier insightful work [2], which proposes a greedy algorithm to tackle the TN-SS problem.
>
> [2] Meraj Hashemizadeh, Michelle Liu, Jacob Miller, and Guillaume Rabusseau. Adaptive learning of tensor network structures. NeurIPS Workshop on Quantum Tensor Networks in Machine Learning, 2020.

---

> > ### Author Response · Authors · 2025-11-23
> > **Re Re Minor Weakness 5**
> >
> > > **Re: Minor Weakness 5 (Dataset Sample Size)**: I myself ran tensor decompositions on the MNIST dataset... still much larger than your time series dataset.
> >
> > We appreciate the reviewer for raising this point, as it allows us to clarify the specific computational challenges faced due to a large sample count of the time-series dataset.
> >
> > While MNIST contains 70,000 samples, these are matrices (greyscale images are order-2 tensors). In contrast, the time-series dataset contains order-5 tensors. The search space for Tensor Network structures grows exponentially with the order.
> >
> > 1. Tensor setting of MNIST: If stacked into a single order-3 tensor ($28 \times 28 \times 70,000$), a single evaluation only calculates the tensor decomposition of an order-3 tensor, making the effective sample size equal to 1.
> > 2. Our time-series dataset: We utilize 142 samples of order-5 tensors. In every evaluation step, the optimization solver must perform 142 tensor network decompositions. Since the computational complexity of the optimization solver scales exponentially with the number of modes, finding the optimal structure for order-5 data is significantly more time-consuming than for order-2 or order-3 data, especially when the number of samples is large.

---

> > > ### Comment · Reviewer_o65j · 2025-11-25
> > >
> > > Thank you for your additional clearifications and changes to the paper, I think they truly improve your work. You said you updated Table 1, but it still seems to be the same, without the mean and std. In general I believe Table 9 should replace it, since it is a fair, but still strong comparison. I understand your argument why SVDinsTN is not applicable to your setting directly. However, I still think that some data driven baseline initialization would strengthen your paper. Since you only need the structure you could run SVDinsTN on all samples and average over the sample axis or just on a single sample, to get a simple baseline structure.
> > >
> > > I raised my score to 6, since you cleared up most of my concerns and showed that tnLLM is actually useful for updating the ranks, a fact that you could place more prominently in the paper. For a stronger recommendation, I still would like to see a stronger initialization baseline.

---

> > > > ### Author Response · Authors · 2025-11-25
> > > >
> > > > Thank you very much for the constructive discussion and for raising your score. We appreciate your time and the detailed feedback that has significantly improved our paper. We will incorporate the detailed comments and ensure that the usefulness of tnLLM for updating ranks is highlighted more prominently in the main text in the camera-ready version.
> > > >
> > > > Regarding SVDinsTN: Thank you for the specific suggestion on how to adapt SVDinsTN. We have started the implementation using the adaptation strategy you proposed. As the discussion period is ending in a few days, we will post the results in a follow-up comment if the experiments finish in time.

---

### Official Review · Reviewer_wwNg · 2025-10-31

**Soundness:** 3
**Presentation:** 3
**Contribution:** 3
**Rating:** 8
**Confidence:** 2

**Summary:**

The paper proposes an LLM-guided tensor network (TN) topology search framework that exploits large language models' ability to parse high-level semantic relationships between mode and domain knowledge. The prompting follows CoT with three prompts—behavior directive, task directive and optimization directive—enable controllable exploration of the discrete TN configuration space.
Performance is quantified on held-out tensor families using the objective function involving compression ratio and structural complexity. The authors report  the competing performance over  three LLM-based baselines, with better convergence rate.

**Strengths:**

* By injecting tensor mode interdependence (e.g., shared physical indices in quantum many-body systems) into prompts, the LLM implicitly learns to suppress topologically invalid contractions, reducing the rate of malformed outputs.
* The experimental results on three types of tensor data across different domains demonstrate the effectiveness of the proposed method.

**Weaknesses:**

* While a scalarized objective (Eq.(1) in the paper) is used, the paper omits raw compression ratio and relative reconstruction error across the Pareto frontier — the de facto standard in TN compression. Reporting only $\mathcal{L}$ obscures trade-offs and hinders comparison with heuristics.
* As shown in Table 2, performance degrade with weak LLMs and may suffer from hallucination issue of LLMs. No robustness analysis is provided.

**Questions:**

* How does the context length affect search performance?
* It would be good to compare the proposed method with non-LLM based methods.

---

> ### Author Response · Authors · 2025-11-21
>
> We appreciate the favorable feedback from the reviewer. Please find our responses to your comments and questions below.
>
> -------
>
> > **Weakness 1**: "While a scalarized objective (Eq.(1) in the paper) is used, the paper omits raw compression ratio and relative reconstruction error across the Pareto frontier — the de facto standard in TN compression. Reporting only  obscures trade-offs and hinders comparison with heuristics."
>
> Thank you for this insightful suggestion. Since TN-SS in compression applications is usually used to find the most compressed TN format given an error threshold, we report here the best compression rate (number of entries in the original tensor/ number of entries in its TN compressed format) achieved by all methods for a given test approximation error threshold (0.02 for Images, 0.01 for Videos and 0.01 for Time-Series dataset). An error threshold of 0.01 can be interpreted as almost a ‘perfect’ reconstruction for standardized tensors. We take 0.02 as the test approximation error threshold for the Images dataset, as none of the methods achieved an approximation error of 0.01. tnLLM is observed to achieve the best compression rates in two of the three datasets.
>
> | | TNLS | TnALE | tnGPS | tnLLM |
> | :--- | :--- | :--- | :--- | :--- |
> | Images | 2.54 | 2.25 | 2.47 | **2.63** |
> | Videos | 12.38 | **12.67** | 11.52 | 11.22 |
> | Time-Series | 1.47 | Failed | 1.49 | **1.60** |
>
> We have also added this table as Table 2 in Section 4 of the revised manuscript.
>
> ---
>
> > **Weakness 2**: "As shown in Table 2, performance degrades with weak LLMs and may suffer from hallucination issues of LLMs. No robustness analysis is provided."
>
> The hallucination observed in weaker models does not affect the performance of tnLLM, as demonstrated in our ablation study. The performance of our proposed framework is robust across different LLMs (GPT-4o, GPT-4.5, GPT-4o-mini, and GPT-3.5) as shown in Table 2 (Table 3 of revised manuscript). This indicates that the robustness of tnLLM stems from the proposed prompting pipeline, rather than any individual LLM. To further demonstrate the robustness of our framework, we also experimented with the open-source DeepSeek-V3-0324 model, which activates only 37B parameters for each token and exhibits comparable performance to all other models across all datasets.
>
> -----
>
> > **Question 1**: "How does the context length affect search performance?"
>
> Benefiting from the efficiency of the proposed tnLLM in terms of number of evaluations, the context length does not limit the search performance. Furthermore, we set the maximum token length for each LLM response to 1000, which leads to fast run-times and consistent performance throughout the optimization process.
>
> ---
>
> > **Question 2**: "It would be good to compare the proposed method with non-LLM based methods."
>
> We believe there may be a slight misunderstanding regarding the baselines. In Table 1 and Figure 7, we compare tnLLM directly against TNLS and TnALE, which are the established SOTA non-LLM sampling-based algorithms. The superiority of our method is demonstrated against these methods along with the LLM-based tnGPS.
>
> ---
>
> We hope to have answered all your questions, and that you will consider increasing your score. If not, please let us know what questions remain and we would be happy to address them.

---

> > ### Comment · Reviewer_wwNg · 2025-11-27
> >
> > Thank you to the authors for their detailed rebuttal. The responses have successfully addressed all of my original concerns.
> >  Given that I have already assigned a high score in the initial review, I remains my score unchanged.

---

### Official Review · Reviewer_fA79 · 2025-11-01

**Soundness:** 3
**Presentation:** 3
**Contribution:** 2
**Rating:** 4
**Confidence:** 2

**Summary:**

This paper addresses the tensor network structure search (TN-SS) problem, which involves identifying optimal tensor network (TN) structures. Current state-of-the-art approaches typically treat TN-SS as a purely numerical optimization or search task, relying on sampling, Bayesian inference, spectral methods, or reinforcement learning, which leads to high computational costs and limited interpretability. To overcome these limitations, the authors propose tnLLM, a domain-aware large language model (LLM)-guided framework that incorporates domain knowledge about tensor modes through tailored prompting. tnLLM enables the LLM to generate promising initial TN structure configurations and iteratively refine them using significantly fewer objective evaluations. Experiments on real-world tensors of orders 3, 4, and 5 demonstrate that tnLLM achieves performance comparable to state-of-the-art sampling-based TN-SS methods while drastically reducing the number of evaluations required.

**Strengths:**

- **S1**. The idea of leveraging LLMs to incorporate domain information into a fundamentally combinatorial search problem (TN-SS) is compelling, which brings a fresh angle to tensor network design.
- **S2**. The experimental results demonstrate substantial reductions in the number of required evaluations (e.g., up to ~78× fewer than a baseline method) while maintaining comparable performance on the target objective.
- **S3**. The ability to provide human-interpretable explanations for the resulting tensor network structures enhances transparency and trustworthiness.
- **S4**. The overall methodology, including the prompt design and search workflow, is clearly presented, and the evaluation spans diverse domains (images, video, and time-series).

**Weaknesses:**

- **W1**. The core technical contribution mainly relies on prompt engineering with LLMs, which limits the methodological novelty and may not be considered a strong theoretical contribution.
- **W2**. The framework lacks theoretical guarantees regarding when the LLM will propose reliable structures and avoid hallucinations, making its behavior difficult to predict.
- **W3**. The experimental evaluation is restricted to relatively small-scale and limited domains (images, videos, and time-series), leaving the generalizability to more complex high-order tensors uncertain.
- **W4**. The explanation capability, while interesting, is not quantitatively evaluated; the correctness and usefulness of the generated explanations remain unclear.

**Questions:**

- **Q1**. The method relies heavily on high-quality domain information. If such information is difficult to obtain, does the advantage of using an LLM diminish significantly?
- **Q2**. The approach requires carefully designed prompts. When extending to new tasks or domains, how should the prompts be adapted or improved to maintain performance?
- **Q3**. Do the authors plan to introduce quantitative measures (e.g., fidelity metrics) to evaluate the accuracy and usefulness of the model-generated explanations?

---

> ### Author Response · Authors · 2025-11-21
>
> We thank the reviewer for highlighting the strengths of our work. We believe there may have been some misunderstanding. Please find our answers to your comments and questions below.
>
> ---
>
> > **W1**. The core technical contribution mainly relies on prompt engineering with LLMs, which limits the methodological novelty and may not be considered a strong theoretical contribution.
>
> We believe this is a misunderstanding and wish to clarify the novelty of our paper. Our work introduces new algorithmic insight into TN-SS by showing for the first time that real-world domain information can be incorporated to solve the TN-SS problem effectively, whereas existing SOTA methods solve TN-SS as a purely numerical optimization problem. The prompting pipeline is the mechanism for integrating domain information, rather than the contribution itself. Additionally, our proposed hybrid algorithm, which not only can utilize domain information, but also has the theoretical guarantees of SOTA TN-SS algorithms, can offer valuable insights into the future development of TN-SS methods.
>
> ---
>
>
> > **W2**. The framework lacks theoretical guarantees regarding when the LLM will propose reliable structures and avoid hallucinations, making its behavior difficult to predict.
>
> To clarify, we do not contribute in terms of theoretical guarantees of TN-SS, as explicitly stated in our paper (Section 4.1 lines 424-425; Appendix A lines 650-651). Our work contributes a new paradigm of TN-SS, which utilises real-world domain information about the tensor data to solve the TN-SS problem.
>
> We also proposed and tested a hybrid algorithm (Section 4.1), which does have theoretical guarantees and can utilize domain information. Additionally, our ablation study (Section 4.2) verifies that the performance of tnLLM is reliable across different LLMs, owing to the carefully designed pipeline. Furthermore, the prompts are carefully structured to define the role of LLM as a domain expert, the specific task, the objective function, and the exact output format. This is crucial for minimizing hallucinations and enabling a fully automated optimization pipeline without the need for any human intervention, as noted in line 289-296.
>
> ---
>
>
> > **W3**. The experimental evaluation is restricted to relatively small-scale and limited domains (images, videos, and time-series), leaving the generalizability to more complex high-order tensors uncertain.
>
>
> We believe there may be a misunderstanding here as the reviewer highlighted the diverse domains as one of the strengths of our paper (**S4**). We thus respectfully disagree with this point. The image and video datasets are the standard datasets used in existing literature of TN-SS methods [1-4] in terms of both size and data domains and are thus used to validate our proposed method. To investigate the generalizability and performance of our method in complex domains, we have also explicitly curated ourselves a custom dataset from the financial domain to ensure that it is not structured and has complex mode relationships which are highly technical, narrow, and domain-specific. This financial time-series tensor dataset is also the largest dataset in terms of sample sizes ever tested in the TN-SS setting, as noted in lines 320-323 and 694-697.
>
> ---
>
> >**W4**. The explanation capability, while interesting, is not quantitatively evaluated; the correctness and usefulness of the generated explanations remain unclear.
>
> Developing systematic quantitative evaluations of the explanations by LLMs is indeed exciting, but is not the focus of our work. In our work, the explanations were already qualitatively verified and primarily serve to mitigate the black-box nature of existing SOTA methods which solve TN-SS as a purely numerical optimization problem, as shown in Figure 6. The transparency offered by the explanations can also be used for ‘debugging’ purposes, enabling practitioners to verify whether the identified TN structure is correct i.e. whether the rank provided by the LLM is justifiable according to the practitioners domain knowledge.
>
> ---
>
>
> [1] C.Li et. al. Alternating local enumeration (TnALE): solving tensor network structure search with fewer evaluations. In Proceedings of the 40th International Conference on Machine Learning, ICML’23. JMLR.org, 2023.
>
> [2] C. Li et. al.  Permutation search of tensor network structures via local sampling. In Proceedings of the 39th International Conference on Machine Learning, pages 13106–13124, Jul 2022.
>
> [3] Y. Zheng et. al. SVDinsTN: A tensor network paradigm for efficient structure search from regularized modeling perspective. In Proceedings of the IEEE/CVF Conference on Computer Vision and Pattern Recognition, 2024.
>
> [4] J. Zeng et. al. tnGPS: Discovering unknown tensor network structure search algorithms via large language models (LLMs). In Proceedings of the Forty-first International Conference on Machine Learning, 2024.

---

> > ### Author Response · Authors · 2025-11-21
> >
> > > **Q1**. The method relies heavily on high-quality domain information. If such information is difficult to obtain, does the advantage of using an LLM diminish significantly?
> >
> > We thank the reviewer for this insightful question about edge cases. In the scenario with a complete absence of domain information, our framework converges less quickly and lacks transparency in the generated TN structures, as noted in Section 4.2. However, we argue that this is an extreme edge case, as most real-world tensors have at least some domain information from the data collection process. In these settings, our tnLLM can utilize the available domain information to guide its initialization and optimization process. This is a significant improvement over current SOTA methods, which typically treat TN-SS as a pure numerical optimization problem and cannot exploit any domain information.
> >
> > ---
> >
> > > **Q2**. The approach requires carefully designed prompts. When extending to new tasks or domains, how should the prompts be adapted or improved to maintain performance?
> >
> > As the system was designed with domain–generalizability in mind, extending our framework to a new task or domain requires modifying only a single component of the prompting pipeline. Specifically, to adapt to a new domain, a practitioner would only need to update the content of the 'Task-directive' prompt with the mode descriptions for that domain, which is an expected requirement for any domain-aware method.
> >
> > ---
> >
> >
> > > **Q3**. Do the authors plan to introduce quantitative measures (e.g., fidelity metrics) to evaluate the accuracy and usefulness of the model-generated explanations?
> >
> > We agree that developing quantitative evaluation metrics for the accuracy of LLM explanations would be an exciting future avenue, but this is not the focus of our contribution. In our work, the explanations were already qualitatively verified and primarily serve to mitigate the black-box nature of existing SOTA methods which solve TN-SS as a purely numerical optimization problem, as shown in Figure 6.
> >
> > ---
> >
> >
> > We hope that our answers have addressed your concerns and have better clarified the novelty of our work. If so, we would like to kindly ask you to increase your score. If not, please let us know which issues remain and we would be happy to address them.

---

> ### Author Response · Authors · 2025-11-27
>
> To further address your concern regarding the **W3** about limited data domains, we have conducted additional experiments on 1) physics, 2) biomedical neuroimaging (MRI), and 3) geospatial hyperspectral data.
>
> For the physics experiment, we use the Darcy Flow dataset from PDEBench [1]. Each tensor has the shape of 128 × 128 × 10, with 10 samples used for training and 10 for testing. For the neuroimaging experiment, we use the BrainWeb MRI dataset [2]. Each tensor has the shape of 217 × 181 × 36 × 2, where 217 is the height, 181 is the width, 36 is the number of slices, and 2 is the modality. We use the normal brain data for training and the brain data with multiple sclerosis for testing. For the geospatial hyperspectral image experiment, we use the datasets provided in [3] and [4]. Each tensor is ensured to share a common shape of 100 × 100 × 80, where each mode represents the width, height, and spectral bands respectively.
>
> The results are summarized in the Table below. The values on the left give the lowest training and corresponding testing objective function values. The values in [square brackets] give the number of evaluations required to first achieve the best training objective function value. As observed, the proposed tnLLM is able to achieve superior or comparable objective values while having significant speed ups in terms of number of evaluations (up to 62x compared to TNLS, 29x compared to TnALE, and 26x compared to tnGPS) in all three additional domains.
>
> | Dataset |   | TNLS | TNALE | TNGPS | TNLLM |
> | :--- | :--- | :--- | :--- | :--- | :--- |
> | **Physics** | Train | -0.57 &nbsp; [196] | -0.56 &nbsp; [162] | -0.55 &nbsp; [$\underline{92}$] | -0.54 &nbsp; [**7**] |
> | | Test | -0.49 | **-0.56** | -0.52 | $\underline{-0.54}$ |
> | **Geospatial Hyperspectral** | Train | -1.45 &nbsp; [205] | -1.43 &nbsp; [151] | -1.43 &nbsp; [$\underline{76}$] | -1.42 &nbsp; [**7**] |
> | | Test | $\underline{-1.33}$ | -1.29 | **-1.35** | $\underline{-1.33}$  |
> | **Biomedical MRI** | Train | -0.46 &nbsp; [310] | -0.45 &nbsp; [146] | -0.41 &nbsp; [$\underline{130}$] | -0.48 &nbsp; [**5**] |
> | | Test | $\underline{-0.54}$ | -0.53 | -0.45 | **-0.55** |
>
> [1] Makoto Takamoto, Timothy Praditia, Raphael Leiteritz, Dan MacKinlay, Francesco Alesiani, Dirk Pflüger, and Mathias Niepert. PDEBench: An Extensive Benchmark for Scientific Machine Learning. In Proceedings of the 36th Conference on Neural Information Processing Systems Datasets and Benchmarks Track, 2022.
>
> [2] Chris A. Cocosco, Vaskev Kollokian, Remi K.-S. Kwan, and Bruce G. Pike. BrainWeb: Online Interface to a 3D MRI Simulated Brain Database. NeuroImage, vol. 5, no. 4, part 2/4, S425, 1997.
>
> [3] Marion F. Baumgardner, Larry L. Biehl, and David A. Landgrebe. 220 Band AVIRIS Hyperspectral Image Data Set: June 12, 1992 Indian Pine Test Site 3. Purdue University Research Repository, 2015.
>
> [4] Fabio Dell'Acqua, Paolo Gamba, Alessandro Ferrari, J. A. Palmason, Jon Atli Benediktsson, and K. Arnason. Exploiting spectral and spatial information in hyperspectral urban data with high resolution. In IEEE Geoscience and Remote Sensing Letters, vol. 1, no. 4. IEEE, 2004.
>
> ---
>
> As the rebuttal deadline approaches, we hope that our answers have resolved your concerns. We would appreciate if you could reconsider your score in light of these clarifications. We are happy to address any further questions you may have.

---

### Author Response · Authors · 2025-12-02
**Author Final Remarks**

We sincerely thank the Area Chair and the Reviewers for their time and constructive feedback. We are pleased that these discussions have led to a score increase from Reviewer **o65j from 2 to 6 (with Reviewers fA79 and pkqP unable to increase their scores in time)**. Our work introduces new algorithmic insights into the Tensor Network Structure Search (TN-SS) problem by showing for the first time that real-world domain information can be incorporated to solve the TN-SS problem effectively, whereas existing SOTA methods solve it as a purely numerical optimization problem. In this way, tnLLM solves the TN-SS problem with superior objective function values, significantly fewer evaluations, and domain-aware explanations.

We provide a summary of the discussions below:

> **1. Experiment Baselines (Addressing Reviewer o65j)**

We had a deep technical discussion with Reviewer o65j regarding the SOTA TN-SS methods and reported further results which have strengthened our proposed method.

- SVDinsTN: We showed mathematically that SVDinsTN is only useful for compressing a **single** tensor, whereas our work focuses on finding **generalizable TN structures for a set of tensors**.

- Baselines: As requested by Reviewer o65j, we tried initializing TNLS and TnALE with random structures and found these to be worse performing. Also, when we reported the number of evaluations of the baselines with early stopping, our method maintained its evaluation efficiency advantage (up to **68x** compared to TNLS, **24x** compared to TnALE, and **53x** compared to tnGPS) with the baseline methods achieving worse objective function values.

- TN-discovery: We provided step-by-step LLM output logs (in Appendix J) and qualitative examples of the TN-discovery process (in Appendix I). Reviewer o65j suggested that these positive results have significantly strengthened our manuscript.

- Outcome: Reviewer o65j acknowledged that we "cleared up most of their concerns" and **"raised their score from 2 to 6"**.


> **2. Generalizability to Additional Domains (Addressing Reviewer pkqP)**

Reviewer pkqP’s primary concern was the diversity of the datasets used in SOTA TN-SS literature. Beyond our curated time-series dataset, we conducted experiments on three additional distinct domains suggested by Reviewer pkqP:

- **Physics**: Darcy Flow dataset from PDEBench
- **Biomedical Neuroimaging**: BrainWeb MRI dataset
- **Geospatial**: Hyperspectral imaging data

The proposed tnLLM achieves superior or comparable objective function values across all three additional domains while demonstrating significant speedups, requiring up to **62x** fewer evaluations compared to TNLS, **29x** fewer than TnALE, and **26x** fewer than tnGPS. These results directly address the concern regarding the generalizability of our method to further domains and are reported in Appendix K of the revised manuscript.

> **3. Theoretical Guarantees (Addressing Reviewer fA79)**

Reviewer fA79 had questions regarding theoretical guarantees of the proposed method.

We explained that the proposed hybrid algorithm **retains the theoretical guarantees** of the SOTA algorithms (e.g., TNLS & TnALE) while benefiting from the acceleration provided by the proposed method. The experiments with three additional data domains also targets Reviewer fA79’s concern about the diversity of domains used in SOTA TN-SS literature.

> **4. Robustness and Metrics (Addressing Reviewer wwNg)**

Reviewer wwNg raised points regarding compression metrics and model robustness.

- Metrics: We reported compression rates for fixed error thresholds similar to previous TN-SS literature, showing that tnLLM achieves the **best compression rates in two of the three datasets**.
- Robustness: We explained that our framework is robust across various LLMs, confirming that the superior performance is driven by our framework design rather than the capability of a specific LLM.
- Outcome: Reviewer wwNg **maintained a score of 8**, stating that we "successfully addressed all of their original concerns".

---

Our work contributes a new paradigm in solving the TN-SS problem, which moves beyond the current purely numerical optimization setting. By incorporating domain information into the TN-SS problem, our proposed method achieves significant speed-ups while obtaining superior or comparable performance in the identified TN structures. In this way, our work provides a fundamental insight to the tensor network community and promises to stimulate further research and development of TN-SS algorithms.

---

### Meta-Review · Area_Chair_MWR4 · 2026-01-09

**Summary:**

The paper uses LLM to incorporate domain knowledge for tensor network search using prompting (rather strange idea, to be honest). It received mixed reviews. Below are concerns.

fA79:
1) The core technical contribution mainly relies on prompt engineering with LLMs, which limits the methodological novelty and may not be considered a strong theoretical contribution.
2) No theoretical guarantess
3) Small scale experiments
4) Explainability needs to to evaluated.

wwNg:
1) Raw compression ratio and relative reconstruction is not reported
2) Performance degradation with weak LLM, no robustness analysis is provided.

pkqP:
1) Limited domains (video and images)

o65j:
1) SOTA is less clear than stated in the paper
2) No analysis of initialization and local optimization independently
3) Comparisons in the experiments are not entirely fair
4) Code in the supplementary is incomplete.

**Reviewer Concerns:**

fA79: Not addressed at all. What is done (prompting 'you are an expert in tensor decompositions... please study domain...) is not a major contribution which somebody misunderstood
pkqP: Addressed (added more data).
wwNg: Addressed
o65j: Addressed

**Reviewer Scores:**

fA79: 4->4
wwNG: 8->8
pkqP: 4->6
o65j: 2-> 8 (happened in the review period)

---

### Decision · Program_Chairs · 2026-01-26

Reject